

# Modeled larval connectivity of a multi-species reef fish and invertebrate assemblage off the coast of Moloka'i, Hawai'i

Emily E. Conklin[1], Anna B. Neuheimer[2,3] and Robert J. Toonen[1]

[1] Hawai'i Institute of Marine Biology, University of Hawai'i at Mānoa, Kāne'ohe, Hawai'i,
[2] Department of Oceanography, University of Hawai'i at Mānoa, Honolulu, Hawai'i,
[3] Aarhus Institute of Advanced Studies (AIAS), Aarhus University, Aarhus, Denmark

## ABSTRACT

We use a novel individual-based model (IBM) to simulate larval dispersal around the island of Moloka'i in the Hawaiian Archipelago. Our model uses ocean current output from the Massachusetts Institute of Technology general circulation model (MITgcm) as well as biological data on four invertebrate and seven fish species of management relevance to produce connectivity maps among sites around the island of Moloka'i. These 11 species span the range of life history characteristics of Hawaiian coral reef species and show different spatial and temporal patterns of connectivity as a result. As expected, the longer the pelagic larval duration (PLD), the greater the proportion of larvae that disperse longer distances, but regardless of PLD (3–270 d) most successful dispersal occurs either over short distances within an island (<30 km) or to adjacent islands (50–125 km). Again, regardless of PLD, around the island of Moloka'i, connectivity tends to be greatest among sites along the same coastline and exchange between northward, southward, eastward and westward-facing shores is limited. Using a graph-theoretic approach to visualize the data, we highlight that the eastern side of the island tends to show the greatest out-degree and betweenness centrality, which indicate important larval sources and connectivity pathways for the rest of the island. The marine protected area surrounding Kalaupapa National Historical Park emerges as a potential source for between-island larval connections, and the west coast of the Park is one of the few regions on Moloka'i that acts as a net larval source across all species. Using this IBM and visualization approach reveals patterns of exchange between habitat regions and highlights critical larval sources and multi-generational pathways to indicate priority areas for marine resource managers.

Subjects Aquaculture, Fisheries and Fish Science, Conservation Biology, Ecology, Marine Biology, Biological Oceanography
Keywords Marine connectivity, Larval dispersal, Marine protected areas, Hawaiian archipelago, Biophysical modeling

# INTRODUCTION

Knowledge of population connectivity is necessary for effective management in marine environments (*Mitarai, Siegel & Winters, 2008*; *Botsford et al., 2009*; *Toonen et al., 2011*). For many species of marine invertebrate and reef fish, dispersal is mostly limited to the

Corresponding author
Emily E. Conklin,
conkline@hawaii.edu

pelagic larval life stage. Therefore, an understanding of larval dispersal patterns is critical for studying population dynamics, connectivity, and conservation in the marine environment (*Jones, Srinivasan & Almany, 2007*; *Lipcius et al., 2008*; *Gaines et al., 2010*; *Toonen et al., 2011*). Many coastal and reef species have a bi-phasic life history in which adults display limited geographic range and high site fidelity, while larvae are pelagic and highly mobile (*Thorson, 1950*; *Scheltema, 1971*; *Strathmann, 1993*; *Marshall et al., 2012*). This life history strategy is not only common to sessile invertebrates such as corals or limpets; many reef fish species have been shown to have a home range of <1 km as adults (*Meyer et al., 2000*; *Meyer, Papastamatiou & Clark, 2010*). Depending on species, the mobile planktonic stage can last from hours to months and has the potential to transport larvae up to hundreds of kilometers away from a site of origin (*Scheltema, 1971*; *Richmond, 1987*; *Shanks, 2009*). Knowledge of larval dispersal patterns can be used to inform effective management, such as marine spatial management strategies that sustain source populations of breeding individuals capable of dispersing offspring to other areas.

Both biological and physical factors impact larval dispersal, although the relative importance of these factors is likely variable among species and sites and remains debated (*Levin, 2006*; *Paris, Chérubin & Cowen, 2007*; *Cowen & Sponaugle, 2009*; *White et al., 2010*). *In situ* data on pelagic larvae are sparse; marine organisms at this life stage are difficult to capture and identify, and are typically found in low densities across large areas of the open ocean (*Clarke, 1991*; *Wren & Kobayashi, 2016*). A variety of genetic and chemistry techniques have therefore been developed to estimate larval connectivity (*Gillanders, 2005*; *Leis, Siebeck & Dixson, 2011*; *Toonen et al., 2011*; *Johnson et al., 2018*). Computer models informed by field and laboratory data have also become a valuable tool for estimating larval dispersal and population connectivity (*Paris, Chérubin & Cowen, 2007*; *Botsford et al., 2009*; *Sponaugle et al., 2012*; *Kough, Paris & Butler IV, 2013*; *Wood et al., 2014*). Individual-based models, or IBMs, can incorporate both biological and physical factors known to influence larval movement. Pelagic larval duration (PLD), for example, is the amount of time a larva spends in the water column before settlement and can vary widely among or even within species (*Toonen & Pawlik, 2001*). PLD affects how far an individual can be successfully transported by ocean currents, and so is expected to directly affect connectivity patterns (*Siegel et al., 2003*; *Shanks, 2009*; *Dawson et al., 2014*). In addition to PLD, adult reproductive strategy and timing (*Carson et al., 2010*; *Portnoy et al., 2013*), fecundity (*Castorani et al., 2017*), larval mortality (*Vikebø et al., 2007*), and larval developmental, morphological, and behavioral characteristics (*Paris, Chérubin & Cowen, 2007*) may all play a role in shaping connectivity patterns. Physical factors such as temperature, bathymetry, and current direction can also substantially influence connectivity (*Cowen & Sponaugle, 2009*). In this study, we incorporated both biotic and abiotic components in an IBM coupled with an oceanographic model to predict fine-scale patterns of larval exchange around the island of Moloka'i in the Hawaiian archipelago.

The main Hawaiian Islands are located in the middle of the North Pacific Subtropical Gyre, and are bordered by the North Hawaiian Ridge current along the northern coasts of the islands and the Hawaii Lee Current along the southern coasts, both of which run east to west and are driven by the prevailing easterly trade winds (*Lumpkin, 1998*; *Friedlander*

*et al., 2005*). The Hawai'i Lee Countercurrent, which runs along the southern perimeter of the chain, flows west to east (*Lumpkin, 1998*). The pattern of mesoscale eddies around the islands is complex and varies seasonally (*Friedlander et al., 2005*; *Vaz et al., 2013*).

Hawaiian marine communities face unprecedented pressures, including coastal development, overexploitation, disease, and increasing temperature and acidification due to climate change (*Smith, 1993*; *Lowe, 1995*; *Coles & Brown, 2003*; *Friedlander et al., 2003*; *Friedlander et al., 2005*; *Aeby, 2006*). Declines in Hawaiian marine resources argue for implementation of a more holistic approach than traditional single-species maximum sustainable yield techniques, which have proven ineffective (*Goodyear, 1996*; *Hilborn, 2011*). There is a general movement toward the use of ecosystem-based management, which requires knowledge of ecosystem structure and connectivity patterns to establish and manage marine spatial planning areas (*Slocombe, 1993*; *Browman et al., 2004*; *Pikitch et al., 2004*; *Arkema, Abramson & Dewsbury, 2006*). Kalaupapa National Historical Park is a federal marine protected area (MPA) located on the north shore of Moloka'i, an island in the Maui Nui complex of the Hawaiian archipelago, that includes submerged lands and waters up to $\frac{1}{4}$ mile offshore (*NOAA, 2009*). At least five IUCN red-listed coral species have been identified within this area (*Kenyon, Maragos & Fenner, 2011*), and in 2010 the Park showed the greatest fish biomass and species diversity out of four Hawaiian National Parks surveyed (*Beets, Brown & Friedlander, 2010*). One of the major benefits expected of MPAs is that the protected waters within the area provide a source of larval spillover to other sites on the island, seeding these areas for commercial, recreational, and subsistence fishing (*McClanahan & Mangi, 2000*; *Halpern & Warner, 2003*; *Lester et al., 2009*).

In this study, we used a Lagrangian particle-tracking IBM (*Wong-Ala et al., 2018*) to simulate larval dispersal around Moloka'i and to estimate the larval exchange among sites at the scale of an individual island. We have parameterized our model with biological data for eleven species covering a breadth of Hawaiian reef species life histories (e.g., habitat preferences, larval behaviors, and pelagic larval durations, Table 1), and of interest to both the local community and resource managers. Our goals were to examine patterns of species-specific connectivity, characterize the location and relative magnitude of connections around Moloka'i, describe sites of potential management relevance, and address the question of whether Kalaupapa National Historical Park provides larval spillover for adjacent sites on Moloka'i, or connections to the adjacent islands of Hawai'i, Maui, O'ahu, Lana'i, and Kaho'olawe.

## METHODS

### Circulation model

We selected the hydrodynamic model MITgcm, which is designed for the study of dynamical processes in the ocean on a horizontal scale. This model solves incompressible Navier–Stokes equations to describe the motion of viscous fluid on a sphere, discretized using a finite-volume technique (*Marshall et al., 1997*). The one-km resolution MITgcm domain for this study extends from 198.2°E to 206°E and from 17°N to 22.2°N, an area that includes the islands of Moloka'i, Maui, Lana'i, Kaho'olawe, O'ahu, and Hawai'i. While Ni'ihau and

**Table 1  Target taxa selected for the study, based on cultural, ecological, and/or economic importance.** PLD = pelagic larval duration. Short dispersers (3–25 day minimum PLD) in white, medium dispersers (30–50 day minimum PLD) in light gray, and long dispersers (140–270 day minimum PLD) in dark gray. Spawn season and timing from traditional ecological knowledge shared by cultural practitioners on the island. Asterisk indicates that congener-level data was used.

| Common name | Scientific name | Spawn type | # of larvae spawned | Spawning day of year | Spawning hour of day | Spawning moon phase | Larval depth (m) | PLD (days) | Habitat |
|---|---|---|---|---|---|---|---|---|---|
| 'Opihi/ Limpet | *Cellana* spp. | Broadcast[1] | 861,300 | 1–60 & 121–181 | – | New | 0–5 | 3–18[1,2] | Intertidal[1] |
| Ko'a/ Cauliflower coral | *Pocillopora meandrina* | Broadcast[3] | 1,671,840 | 91–151 | 07:15–08:00 | Full | 0–5[4] | 5–90*[5] | Reef |
| He'e/ Octopus | *Octopus cyanea* | Benthic[6] | 1,392,096 | 1–360 | – | – | 50–100 | 21[6] | Reef, rubble[7] |
| Moi/ Pacific threadfin | *Polydactylus sexfilis* | Broadcast | 1,004,640 | 152–243 | – | – | 50–100[8] | 25[9] | Sand[10] |
| Uhu uliuli/ Spectacled parrotfish | *Chlorurus perspicillatus* | Broadcast | 1,404,792 | 152–212 | – | – | 0–120*[11] | 30*[12] | Reef[10] |
| Uhu palukaluka/ Reddlip parrotfish | *Scarus rubroviolaceus* | Broadcast | 1,404,792 | 152–212 | – | – | 0–120*[11] | 30*[12] | Rock, reef[10] |
| Kumu/ Whitesaddle Goatfish | *Parupeneus porphyreus* | Broadcast | 1,071,252 | 32–90 | – | – | 0–50*[11] | 41–56*[12] | Sand, rock, reef[10] |
| Kole/ Spotted surgeonfish | *Ctenochaetus strigosus* | Broadcast | 1,177,200 | 60–120 | – | – | 50–100[11] | 50*[12] | Rock, reef, rubble[10] |
| 'Ōmilu/ Bluefin trevally | *Caranx melampygus* | Broadcast | 1,310,616 | 121–243 | – | – | 0–80*[11] | 140*[13,14] | Sand, reef[15] |
| Ulua/ Giant trevally | *Caranx ignoblis* | Broadcast | 1,151,040 | 152–243 | – | Full | 0–80*[11] | 140[13,14] | Sand, rock, reef[15] |
| Ula/ Spiny lobster | *Panulirus* spp. | Benthic[16] | 1,573,248 | 152–243 | – | – | 50–100[16] | 270[17] | Rock, pavement[16] |

**Notes.**

(1) (*Bird et al., 2007*); (2) (*Corpuz, 1983*); (3) (*Schmidt-Roach et al., 2012*); (4) (*Storlazzi Brown & Field, 2006*); (5) (*Richmond, 1987*); (6) (*Heukelem, 1973*); (7) (*Forsythe & Hanlon, 1997*); (8) (*Boehlert, Watson & Sun, 1992*); (9) (*Callan et al., 2012*); (10) (*Mundy, 2005*); (11) (*Boehlert & Mundy, 1996*); (12) (*Luiz et al., 2013*); (13) (*Sudekum, 1984*); (14) (*Longenecker, Langston & Barrett, 2008*); (15) (*DeMartini, DiNardo & Williams, 2003*); (16) (*Polovina & Moffitt, 1995*); (17) (*Phillips et al., 2006*).

southern Kaua'i also fall within the domain, we discarded connectivity to these islands because they lie within the 0.5° boundary zone of the current model. Boundary conditions are enforced over 20 grid points on all sides of the model domain. Vertically, the model is divided into 50 layers that increase in thickness with depth, from five m at the surface (0.0–5.0 m) to 510 m at the base (4,470 –4,980 m). Model variables were initialized using the output of a Hybrid Coordinate Ocean Model (HYCOM) at a horizontal resolution of 0.04° (∼four km) configured for the main Hawaiian Islands, using the General Bathymetric Chart of the Oceans database (GEBCO, 1/60°) (*Jia et al., 2011*).
The simulation runs from March 31st, 2011 to July 30th, 2013 with a temporal resolution of 24 h and shows seasonal eddies as well as persistent mesoscale features (Fig. S1). We do not include tides in the model due to temporal resolution. Our model period represents a neutral ocean state; no El Niño or La Niña events occurred during this time period. To ground-truth the circulation model, we compared surface current output to real-time trajectories of surface drifters from the GDP Drifter Data Assembly Center (Fig. S2) (*Elipot et al., 2016*), as well as other current models of the area (*Wren et al., 2016*; *Storlazzi et al., 2017*).

## Biological model

To simulate larval dispersal, we used a modified version of the *Wong-Ala et al. (2018)* IBM, a 3D Lagrangian particle-tracking model written in the R programming language (*R Core Team, 2017*). The model takes the aforementioned MITgcm current products as input, as well as shoreline shapefiles extracted from the full resolution NOAA Global Self-consistent Hierarchical High-resolution Geography database, v2.3.0 (*Wessel & Smith, 1996*). Our model included 65 land masses within the geographic domain, the largest being the island of Hawai'i and the smallest being Pu'uki'i Island, a 1.5-acre islet off the eastern coast of Maui. To model depth, we used the one arc-minute-resolution ETOPO1 bathymetry, extracted using the R package 'marmap' (*Amante & Eakins, 2009*; *Pante & Simon-Bouhet, 2013*).

Each species was simulated with a separate model run. Larvae were modeled from spawning to settlement and were transported at each timestep ($t = 2$ h) by advection-diffusion transport. This transport consisted of (1) advective displacement caused by water flow, consisting of east ($u$) and north ($v$) velocities read from daily MITgcm files, and (2) additional random-walk displacement, using a diffusion constant of 0.2 m$^2$/s$^{-1}$ (*Lowe et al., 2009*). Vertical velocities ($w$) were not implemented by the model; details of vertical larval movement are described below. Advection was interpolated between data points at each timestep using an Eulerian 2D barycentric interpolation method. We chose this implementation over a more computationally intensive interpolation method (i.e., fourth-order Runge–Kutta) because we did not observe a difference at this timestep length. Biological processes modeled include PLD, reproduction timing and location, mortality, and ontogenetic changes in vertical distribution; these qualities were parameterized via species-specific data obtained from previous studies and from the local fishing and management community (Table 1).

Larvae were released from habitat-specific spawning sites and were considered settled if they fell within a roughly one-km contour around reef or intertidal habitat at the end of their pelagic larval duration. Distance from habitat was used rather than water depth because Penguin Bank, a relatively shallow bank to the southwest of Moloka'i, does not represent suitable habitat for reef-associated species. PLD for each larva was a randomly assigned value between the minimum and maximum PLD for that species, and larvae were removed from the model if they had reached their PLD and were not within a settlement zone. No data on pre-competency period were available for our study species, so this parameter was not included. Mortality rates were calculated as larval half-lives; e.g., one-half of all larvae were assumed to have survived at one-half of the maximum PLD for that species (following

*Holstein, Paris & Mumby, 2014*). Since our focus was on potential connectivity pathways, reproductive rates were calibrated to allow for saturation of possible settlement sites, equating from ~900,000 to ~1,7000,000 larvae released depending on species. Fecundity was therefore derived not from biological data, but from computational minimums.

Development, and resulting ontogenetic changes in behavior, is specific to the life history of each species. Broadcast-spawning species with weakly-swimming larvae (*P. meandrina* and *Cellana* spp., Table 1) were transported as passive particles randomly distributed between 0–5 m depth (*Storlazzi, Brown & Field, 2006*). Previous studies have demonstrated that fish larvae have a high degree of control over their vertical position in the water column (*Irisson et al., 2010*; *Huebert, Cowen & Sponaugle, 2011*). Therefore, we modeled broadcast-spawning fish species with a 24-hour passive buoyant phase to simulate eggs pre-hatch, followed by a pelagic larval phase with a species-specific depth distribution. For *C. ignoblis, C. melampygus, P. porphyreus, C. perspicillatus,* and *S. rubrioviolaceus,* we used genus-level depth distributions (Fig. S3) obtained from the 1996 NOAA ichthyoplankton vertical distributions data report (*Boehlert & Mundy, 1996*). *P. sexfilis* and *C. strigosus* larvae were randomly distributed between 50–100 m (*Boehlert, Watson & Sun, 1992*). Benthic brooding species (*O. cyanea* and *Panulirus* spp.) do not have a passive buoyant phase, and thus were released as larvae randomly distributed between 50–100 m. At each time step, a larva's depth was checked against bathymetry, and was assigned to the nearest available layer if the species-specific depth was not available at these coordinates.

For data-poor species, we used congener-level estimates for PLD (see Table 1). For example, there is no estimate of larval duration for *Caranx* species, but in Hawai'i peak spawning occurs in May–July and peak recruitment in August–December (*Sudekum, 1984*; *Longenecker, Langston & Barrett, 2008*). In consultation with resource managers and community members, a PLD of 140 days was chosen pending future data that indicates a more accurate pelagic period.

## Habitat selection

Spawning sites were generated using data from published literature and modified after input from Native Hawaiian cultural practitioners and the Moloka'i fishing community (Fig. 1). Species-specific habitat suitability was inferred from the 2013–2016 Marine Biogeographic Assessment of the Main Hawaiian Islands (*Costa & Kendall, 2016*). We designated coral habitat as areas with 5–90% coral cover, or ≥1 site-specific coral species richness, for a total of 127 spawning sites on Moloka'i. Habitat for reef invertebrates followed coral habitat, with additional sites added after community feedback for a total of 136 sites. Areas with a predicted reef fish biomass of 58–1,288 g/m$^2$ were designated as reef fish habitat (*Stamoulis et al., 2016*), for a total of 109 spawning sites. Sand habitat was designated as 90–100% uncolonized for a total of 115 sites. Intertidal habitat was designated as any rocky shoreline area not covered by sand or mud, for a total of 87 sites. Number of adults was assumed equal at all sites. For regional analysis, we pooled sites into groups of two to 11 sites based on benthic habitat and surrounding geography (Fig. 1A). Adjacent sites were grouped if they shared the same benthic habitat classification and prevailing wave direction, and/or were part of the same reef tract.

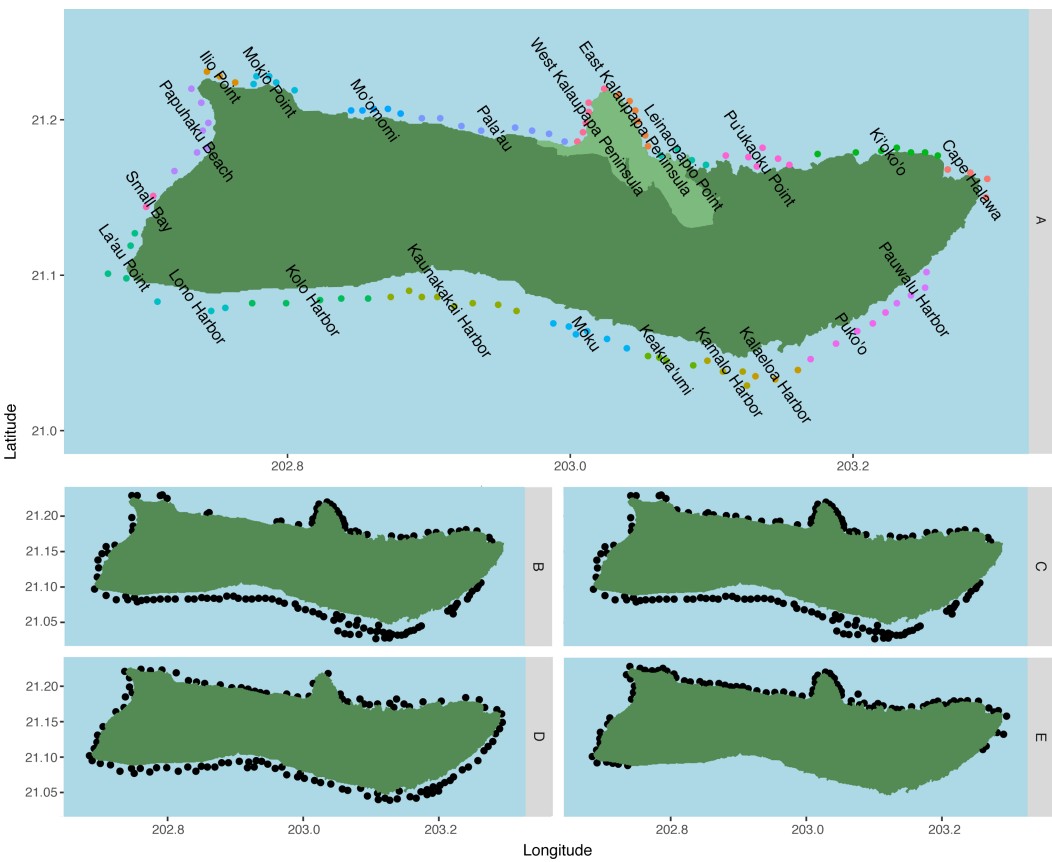

**Figure 1** **Spawning sites used in the model by species.** (A) *C. perspicillatus*, *S. rubroviolaceus*, *P. porphyreus*, *C. strigosus*, *C. ignoblis*, and *C. melampygus*, *n* = 109; (B) *P. meandrina*, *n* = 129; (C) *O. cyanea* and *Panulirus* spp., *n* = 136; (D) *P. sexfilis*, *n* = 115; and (E) *Cellana* spp., *n* = 87. Region names are displayed over associated spawning sites for fish species in (A). Regions are made up of two to 11 sites, grouped based on coastal geography and surrounding benthic habitat, and are designated in (A) by adjacent colored dots. Kalaupapa National Historical Park is highlighted in light green in (A).

## Source–sink dynamics and local retention

Dispersal distance was measured via the *distm* function in the R package 'geosphere', which calculates distance between geographical points via the Haversine formula (*Hijmans, 2016*). This distance, measured between spawn and settlement locations, was used to calculate dispersal kernels to examine and compare species-specific distributions. We also measured local retention, or the percentage of successful settlers from a site that were retained at that site (i.e., settlers at site *A* that originated from site *A*/total successful settlers that originated from site *A*). To estimate the role of specific sites around Moloka'i, we also calculated a source–sink index for each species (*Holstein, Paris & Mumby, 2014*; *Wren et al., 2016*). This index defines sites as either a source, in which a site's successful export to other sites is greater than its import, or a sink, in which import from other sites is greater than successful export. It is calculated by dividing the difference between number of successfully exported and imported larvae by the sum of all successfully exported and imported larvae. A value <0 indicates that a site acts as a net sink, while a value >0 indicates that a site acts as a
net source. While we measured successful dispersal to adjacent islands, we did not spawn larvae from them, and therefore these islands represent exogenous sinks. For this reason, settlement to other islands was not included in source–sink index calculations.

We also calculated settlement proportion between different regions for each species (*Calabrese & Fagan, 2004*). We calculated the forward settlement proportion, i.e., the proportion of settlers from a specific settlement site ($s$) originating from an observed origin site ($o$), by scaling the number of successful settlers from site $o$ settling at site $s$ to all successful settlers originating from site $o$. Forward proportion can be represented as $P_{so} = S_{os}/\sum S_o$. We also calculated rearward settlement proportion, or the proportion of settlers from a specific origin site ($o$) observed at settlement site ($s$), by scaling the number of settlers observed at site $s$ originating from site $o$ to all settlers observed at site $s$. The rearward proportion can be represented as $P_{os} = S_{os}/\sum S_s$.

## Graph-theoretic analysis

To quantify connections between sites, we applied graph theory to population connectivity (*Treml et al., 2008*; *Holstein, Paris & Mumby, 2014*). Graph theoretic analysis is highly scalable and can be used to examine fine-scale networks between reef sites up to broad-scale analyses between islands or archipelagos, mapping to both local and regional management needs. It also allows for both network- and site-specific metrics, enabling the comparison of connectivity between species and habitat sites as well as highlighting potential multi-generational dispersal corridors. Graph theory also provides a powerful tool for spatial visualization, allowing for rapid, intuitive communication of connectivity results to researchers, managers, and the public alike. This type of analysis can be used to model pairwise relationships between spatial data points by breaking down individual-based output into a series of nodes (habitat sites) and edges (directed connections between habitat sites). We then used these nodes and edges to examine the relative importance of each site and dispersal pathway to the greater pattern of connectivity around Moloka'i, as well as differences in connectivity patterns between species (*Treml et al., 2008*; *Holstein, Paris & Mumby, 2014*). We used the R package 'igraph' to examine several measures of within-island connectivity (*Csardi & Nepusz, 2006*). Edge density, or the proportion of realized edges out of all possible edges, is a multi-site measure of connectivity. Areas with a higher edge density have more direct connections between habitat sites, and thus are more strongly connected. We measured edge density along and between the north, south, east, and west coasts of Moloka'i to examine possible population structure and degree of exchange among the marine resources of local communities.

The distribution of shortest path length is also informative for comparing overall connectivity. In graph theory, a shortest path is the minimum number of steps needed to connect two sites. For example, two sites that exchange larvae in either direction are connected by a shortest path of one, whereas if they both share larvae with an intermediate site but not with each other, they are connected by a shortest path of two. In a biological context, shortest path can correspond to number of generations needed for exchange: sites with a shortest path of two require two generations to make a connection. Average shortest path, therefore, is a descriptive statistic to estimate connectivity of a network. If two sites

are unconnected, it is possible to have infinite-length shortest paths; here, these infinite values were noted but not included in final analyses.

Networks can also be broken in connected components (*Csardi & Nepusz, 2006*). A weakly connected component (WCC) is a subgraph in which all nodes are not reachable by other nodes. A network split into multiple WCCs indicates separate populations that do not exchange any individuals, and a large number of WCCs indicates a low degree of island-wide connectivity. A strongly connected component (SCC) is a subgraph in which all nodes are directly connected and indicates a high degree of connectivity. A region with many small SCCs can indicate high local connectivity but low island-wide connectivity. Furthermore, component analysis can identify cut nodes, or nodes that, if removed, break a network into multiple WCCs. Pinpointing these cut nodes can identify potential important sites for preserving a population's connectivity, and could inform predictions about the impact of site loss (e.g., a large-scale coral bleaching event) on overall connectivity.

On a regional scale, it is important to note which sites are exporting larvae to, or importing larvae from, other sites. To this end, we examined in-degree and out-degree for each region. In-degree refers to the number of inward-directed edges to a specific node, or how many other sites provide larvae into site '*A*'. Out-degree refers to the number of outward-directed edges from a specific node, or how many sites receive larvae from site '*A*'. Habitat sites with a high out-degree seed a large number of other sites, and indicate potentially important larval sources, while habitat sites with a low in-degree rely on a limited number of larval sources and may therefore be dependent on connections with these few other sites to maintain population size. Finally, betweenness centrality (BC) refers to the number of shortest paths that pass through a given node, and may therefore indicate connectivity pathways or 'chokepoints' that are important to overall connectivity on a multigenerational timescale. BC was weighted with the proportion of dispersal as described in the preceding section. We calculated in-degree, out-degree, and weighted betweenness centrality for each region in the network for each species.

As with the source–sink index, we did not include sites on islands other than Moloka'i in our calculations of edge density, shortest paths, connected components, cut nodes, in- and out-degree, or betweenness centrality in order to focus on within-island patterns of connectivity.

## RESULTS

### Effects of biological parameters on fine-scale connectivity patterns

The species-specific parameters that were available to parameterize the dispersal models substantially influenced final output (Fig. 2). The proportion of successful settlers (either to Moloka'i or to neighboring islands) varied widely by species, from 2% (*Panulirus* spp.) to 25% (*Cellana* spp.). Minimum pelagic duration and settlement success were negatively correlated (e.g., an estimated $-0.79$ Pearson correlation coefficient). Species modeled with batch spawning at a specific moon phase and/or time of day (*Cellana* spp., *P. meandrina*, and *C. ignoblis*) displayed slightly higher settlement success than similar species modeled with constant spawning over specific months. On a smaller scale, we also examined average

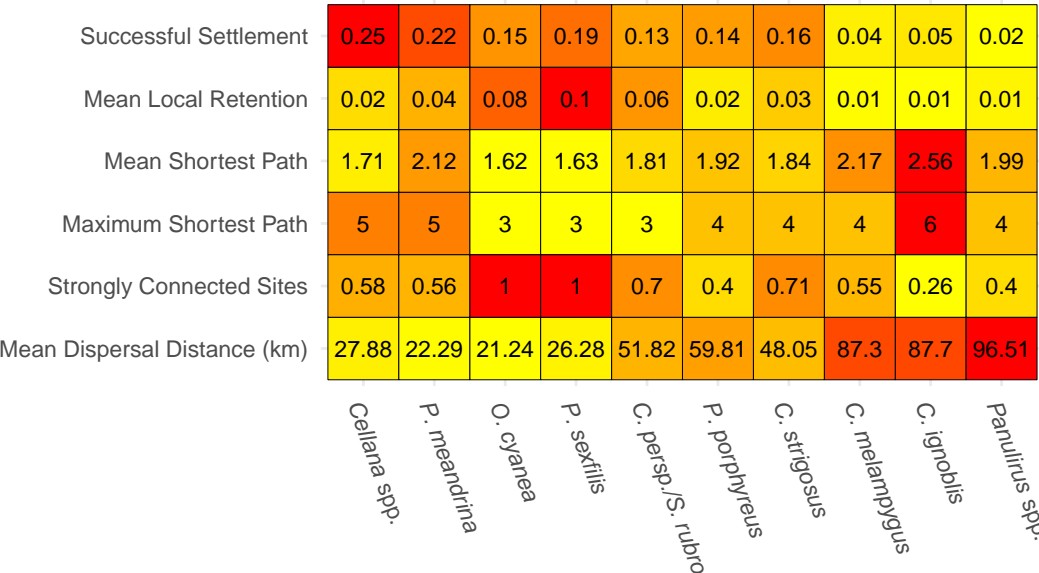

| | Cellana spp. | P. meandrina | O. cyanea | P. sexfilis | C. persp./S. rubro. | P. porphyreus | C. strigosus | C. melampygus | C. ignobilis | Panulirus spp. |
|---|---|---|---|---|---|---|---|---|---|---|
| Successful Settlement | 0.25 | 0.22 | 0.15 | 0.19 | 0.13 | 0.14 | 0.16 | 0.04 | 0.05 | 0.02 |
| Mean Local Retention | 0.02 | 0.04 | 0.08 | 0.1 | 0.06 | 0.02 | 0.03 | 0.01 | 0.01 | 0.01 |
| Mean Shortest Path | 1.71 | 2.12 | 1.62 | 1.63 | 1.81 | 1.92 | 1.84 | 2.17 | 2.56 | 1.99 |
| Maximum Shortest Path | 5 | 5 | 3 | 3 | 3 | 4 | 4 | 4 | 6 | 4 |
| Strongly Connected Sites | 0.58 | 0.56 | 1 | 1 | 0.7 | 0.4 | 0.71 | 0.55 | 0.26 | 0.4 |
| Mean Dispersal Distance (km) | 27.88 | 22.29 | 21.24 | 26.28 | 51.82 | 59.81 | 48.05 | 87.3 | 87.7 | 96.51 |

**Figure 2** **Summary statistics for each species network.** Summary statistics are displayed in order of increasing minimum pelagic larval duration from left to right. Heatmap colors are based on normalized values from 0–1 for each analysis. Successful settlement refers to the proportion of larvae settled out of the total number of larvae spawned. Local retention is measured as the proportion of larvae spawned from a site that settle at the same site. Shortest path is measured as the minimum number of steps needed to connect two sites. Strongly connected sites refers to the proportion of sites in a network that belong to a strongly connected component. Mean dispersal distance is measured in kilometers from spawn site to settlement site.

site-scale local retention, comparing only retention to the spawning site versus other sites on Molokaʻi (Fig. 2). Local retention was lowest for *Caranx* spp. (<1%) and highest for *O. cyanea* and *P. sexfilis* (8.1% and 10%, respectively).

We measured network-wide connectivity via distribution of shortest paths, or the minimum number of steps between a given two nodes in a network, only including sites on Molokaʻi (Fig. 2). *O. cyanea* and *P. sexfilis* showed the smallest shortest paths overall, meaning that on average, it would take fewer generations for these species to demographically bridge any given pair of sites. Using maximum shortest path, it could take these species three generations at most to connect sites. *Cellana* spp. and *P. meandrina*, by comparison, could take as many as five generations. Other medium- and long-dispersing species showed relatively equivalent shortest-path distributions, with trevally species showing the highest mean path length and therefore the lowest island-scale connectivity.

The number and size of weakly-connected and strongly-connected components in a network is also an informative measure of connectivity (Fig. 2). No species in our study group was broken into multiple weakly-connected components; however, there were species-specific patterns of strongly connected sites. *O. cyanea* and *P. sexfilis* were the most strongly connected, with all sites in the network falling into a single SCC. *Cellana* spp. and *P. meandrina* each had approximately 60% of sites included in a SCC, but both show fragmentation with seven and six SCCs respectively, ranging in size from two to 22

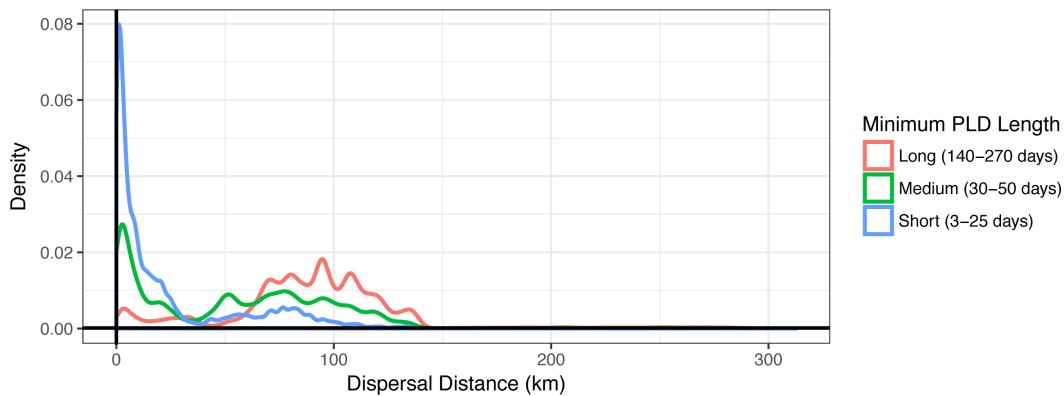

**Figure 3 Dispersal distance density kernels.** Dispersal distance is combined across species by minimum pelagic larval duration (PLD) length in days (short, medium, or long). Most short dispersers settle close to home, while few long dispersers are retained at or near their spawning sites.

sites. This SCC pattern suggests low global connectivity but high local connectivity for these species. Medium and long dispersers showed larger connected components; 70% of parrotfish sites fell within two SCCs; 40% of *P. porphyreus* sites fell within two SCCs; 70% of *C. strigosus* sites, 55% of *C. melampygus* sites, and 40% of *Panulirus* sites fell within a single SCC. In contrast, only 26% of *C. ignoblis* sites fell within a single SCC. It is also important to note that the lower connectivity scores observed in long-dispersing species likely reflect a larger scale of connectivity. Species with a shorter PLD are highly connected at reef and island levels but may show weaker connections between islands. Species with a longer PLD, such as trevally or spiny lobster, are likely more highly connected at inter-island scales which reflects the lower connectivity scores per island shown here.

Minimum PLD was positively correlated with mean dispersal distance (e.g., an estimated 0.88 Pearson correlation coefficient with minimum pelagic duration $\log_e$-transformed to linearize the relationship), and dispersal kernels differed between species that are short dispersers (3–25 days), medium dispersers (30–50 days), or long dispersers (140–270 days) (Fig. 3). Short dispersers travelled a mean distance of 24.06 ± 31.33 km, medium dispersers travelled a mean distance of 52.71 ± 40.37 km, and long dispersers travelled the farthest, at a mean of 89.41 ± 41.43 km. However, regardless of PLD, there were essentially two peaks of mean dispersal: a short-distance peak of <30 km, and a long-distance peak of roughly 50–125 km (Fig. 3). The short-distance peak largely represents larvae that settle back to Moloka'i, while the long-distance peak largely represents settlement to other islands; the low point between them corresponds to deep-water channels between islands, i.e., unsuitable habitat for settlement. Median dispersal distance for short dispersers was substantially less than the mean at 8.85 km, indicating that most of these larvae settled relatively close to their spawning sites, with rare long-distance dispersal events bringing up the average. Median distance for medium (54.22 km) and long (91.57 km) dispersers was closer to the mean, indicating more even distance distributions and thus a higher probability of long-distance dispersal for these species. Maximum dispersal distance varied

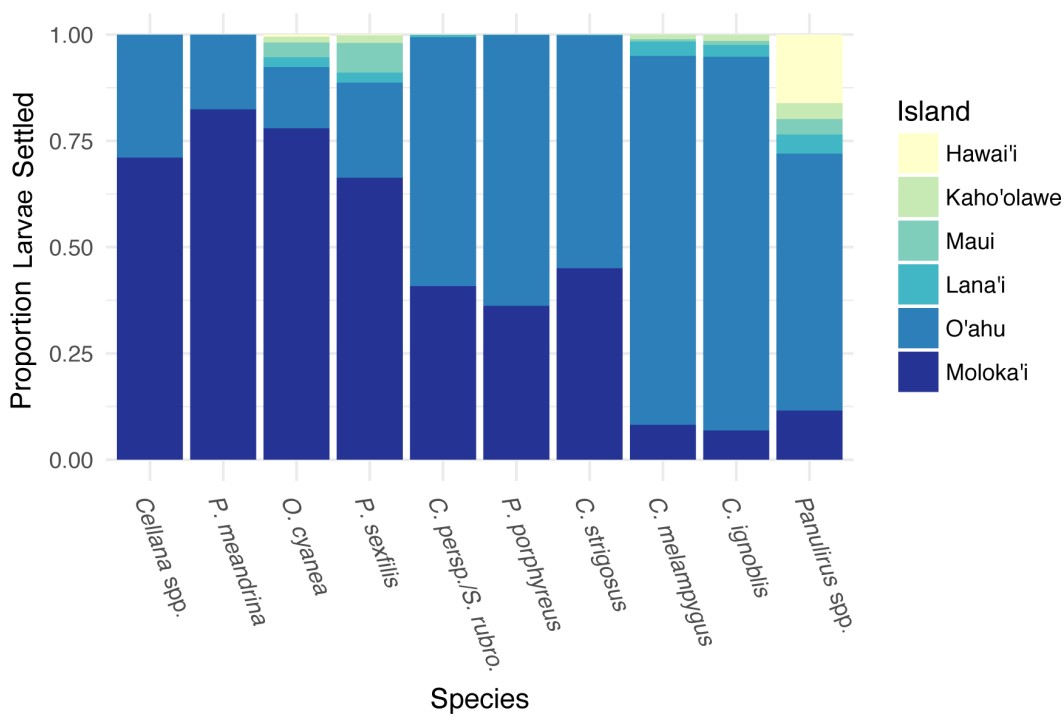

**Figure 4   Forward settlement from Moloka'i to other islands.** Proportion of simulated larvae settled to each island from Moloka'i by species, organized in order of increasing minimum pelagic larval duration from left to right.

between ~150–180 km depending on species, except for the spiny lobster *Panulirus* spp., with a PLD of 270 d and a maximum dispersal distance of approximately 300 km.

## Settlement to Moloka'i and other islands in the archipelago

Different species showed different forward settlement proportion to adjacent islands (Fig. 4), although every species in the study group successfully settled back to Moloka'i. *P. meandrina* showed the highest percentage of island-scale local retention (82%), while *C. ignoblis* showed the lowest (7%). An average of 74% of larvae from short-dispersing species settled back to Moloka'i, as compared to an average of 41% of medium dispersers and 9% of long dispersers. A large proportion of larvae also settled to O'ahu, with longer PLDs resulting in greater proportions, ranging from 14% of *O. cyanea* to 88% of *C. ignoblis*. Moloka'i and O'ahu were the most commonly settled islands by percentage. Overall, settlement from Moloka'i to Lana'i, Maui, Kaho'olawe, and Hawai'i was somewhat lower. Larvae of every species settled to Lana'i, and settlement to this island made up less than 5% of settled larvae across all species. Likewise, settlement to Maui made up less than 7% of settlement across species, with *P. meandrina* as the only species that had no successful paths from Moloka'i to Maui. Settlement to Kaho'olawe and Hawai'i was less common, with the exception of *Panulirus* spp., which had 16% of all settled larvae on Hawai'i.

We also examined coast-specific patterns of rearward settlement proportion to other islands, discarding connections with a very low proportion of larvae (<0.1% of total larvae

of that species settling to other islands). Averaged across species, 83% of larvae settling to Oʻahu from Molokaʻi were spawned on the north shore of Molokaʻi, with 12% spawned on the west shore (Fig. S4). Spawning sites on the east and south shores contributed <5% of all larvae settling to Oʻahu from Molokaʻi. The east and south shores of Molokaʻi had the highest average percentage of larvae settling to Lanaʻi from Molokaʻi, at 78% and 20% respectively, and to Kahoʻolawe from Molokaʻi at 63% and 34%. Of the species that settled to Maui from Molokaʻi, on average most were spawned on the east (53%) or north (39%) shores, as were the species that settled to Hawaiʻi Island from Molokaʻi (22% east, 76% north). These patterns indicate that multiple coasts of Molokaʻi have the potential to export larvae to neighboring islands.

Temporal settlement profiles also varied by species (Fig. 5). Species modeled with moon-phase spawning and relatively short settlement windows (*Cellana* spp. and *C. ignoblis*) were characterized by discrete settlement pulses, whereas other species showed settlement over a broader period of time. Some species also showed distinctive patterns of settlement to other islands; our model suggests specific windows when long-distance dispersal is possible, as well as times of year when local retention is maximized (Fig. 5).

## Regional patterns of connectivity in Molokaʻi coastal waters

Within Molokaʻi, our model predicts that coast-specific population structure is likely; averaged across all species, 84% of individuals settled back to the same coast on which they were spawned rather than a different coast on Molokaʻi. Excluding connections with a very low proportion of larvae (<0.1% of total larvae of that species that settled to Molokaʻi), we found that the proportion of coast-scale local retention was generally higher than dispersal to another coast, with the exception of the west coast (Fig. 6A). The north and south coasts had a high degree of local retention in every species except for the long-dispersing *Panulirus* spp., and the east coast also had high local retention overall. Between coasts, a high proportion of larvae that spawned on the west coast settled on the north coast, and a lesser amount of larvae were exchanged from the east to south and from the north to east. With a few species-specific exceptions, larval exchange between other coasts of Molokaʻi was negligible.

We also calculated edge density, including all connections between coasts on Molokaʻi regardless of settlement proportion (Fig. 6B). The eastern coast was particularly well-connected, with an edge density between 0.14 and 0.44, depending on the species. The southern shore showed high edge density for short and medium dispersers (0.16–0.39) but low for long dispersers (<0.005). The north shore also showed relatively high edge density (0.20 on average), although these values were smaller for long dispersers. The west coast showed very low edge density, with the exceptions of *O. cyanea* (0.37) and *P. sexfilis* (0.13). Virtually all networks that included two coasts showed lower edge density. One exception was the east/south shore network, which had an edge density of 0.10–0.65 except for *Cellana* spp. Across species, edge density between the south and west coasts was 0.12 on average, and between the east and west coasts was 0.04 on average. Edge density between north and south coasts was particularly low for all species (<0.05), a divide that was especially distinct in *Cellana* spp. and *P. meandrina,* which showed zero realized connections between these

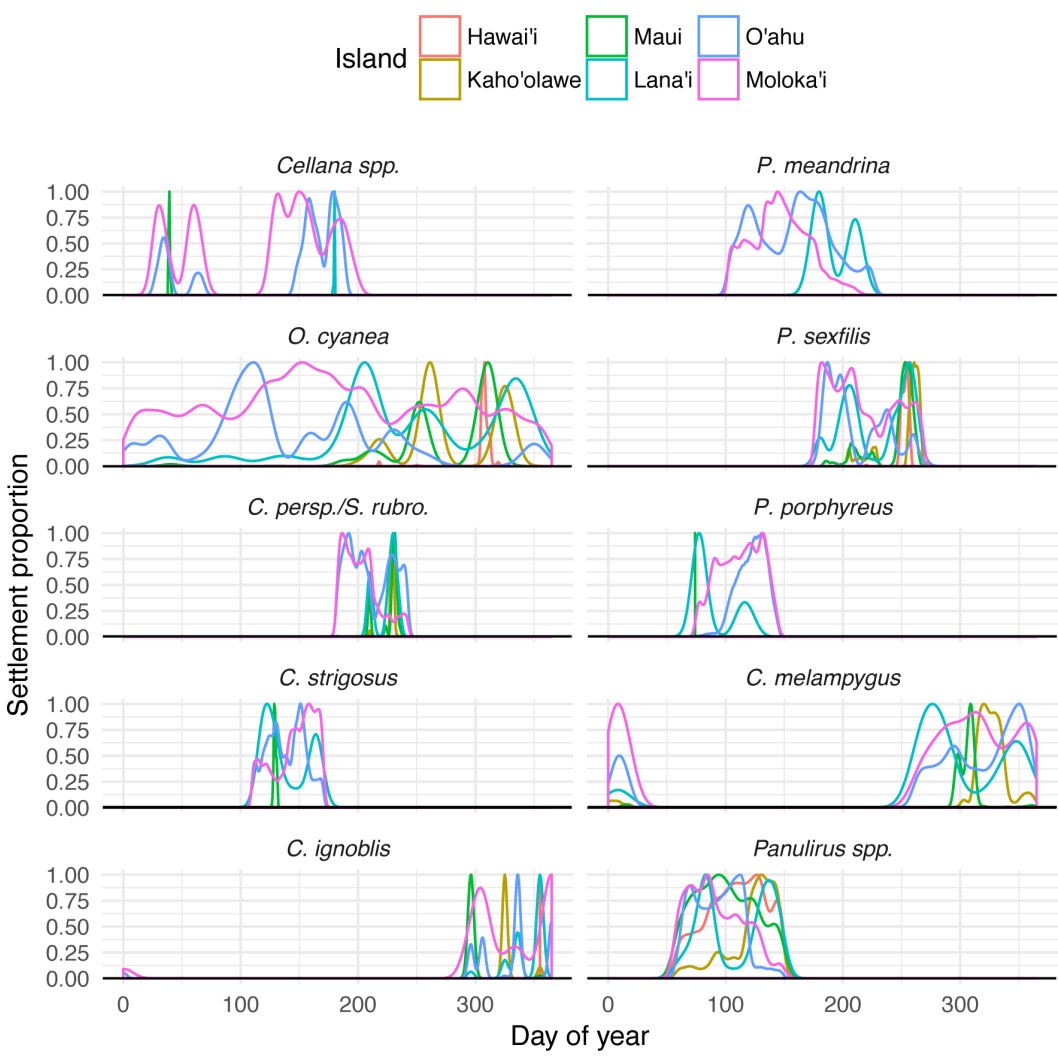

**Figure 5 Species-specific temporal recruitment patterns.** Proportion densities of settlement to specific islands from Molokaʻi based on day of year settled, by species. Rare dispersal events (e.g., Maui or Lanaʻi for *Cellana* spp.) appear as narrow spikes, while broad distributions generally indicate more common settlement pathways.

coasts. Although northern and southern populations are potentially weakly connected by sites along the eastern ( *P. meandrina*) or western (*Cellana* spp.) shores, our model predicts very little, if any, demographic connectivity.

To explore patterns of connectivity on a finer scale, we pooled sites into regions (as defined in Fig. 1) in order to analyze relationships between these regions. Arranging model output into node-edge networks clarified pathways and regions of note, and revealed several patterns which did not follow simple predictions based on PLD (Fig. 7). *Cellana* spp. and *P. meandrina* showed the most fragmentation, with several SCCs and low connectivity between coasts. Connectivity was highest in *O. cyanea* and *P. sexfilis*, which had a single SCC containing all regions. Medium and long dispersers generally

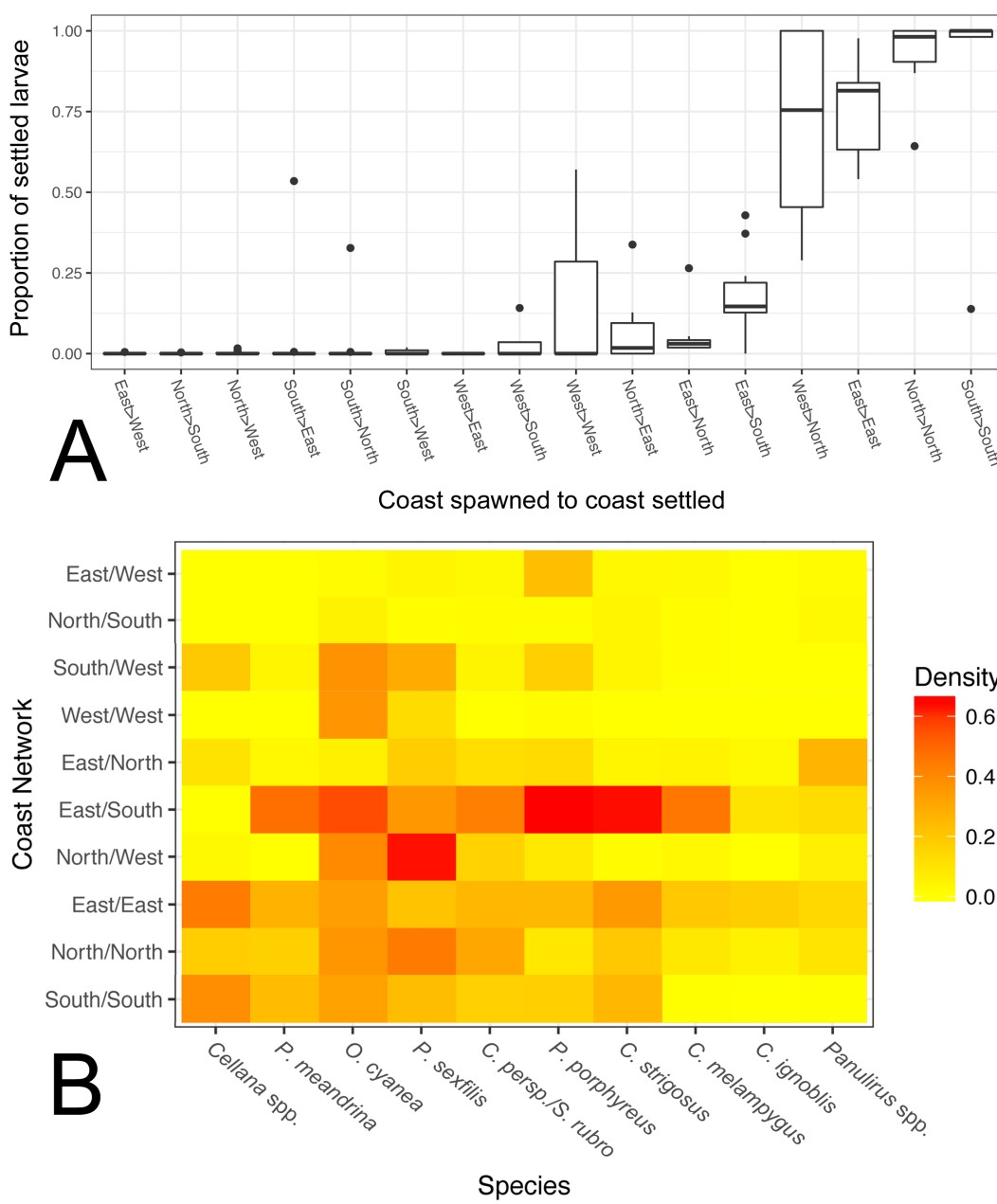

**Figure 6 Coast-by-coast patterns of connectivity on Moloka'i.** (A) Average rearward settlement proportion by species per pair of coastlines, calculated by the number of larvae settling at site *s* from site *o* divided by all settled larvae at site *s*. Directional coastline pairs (Spawn > Settlement) are ordered from left to right by increasing median settlement proportion. (B) Heatmap of edge density for coast-specific networks by species. Density is calculated by the number of all realized paths out of total possible paths, disregarding directionality.

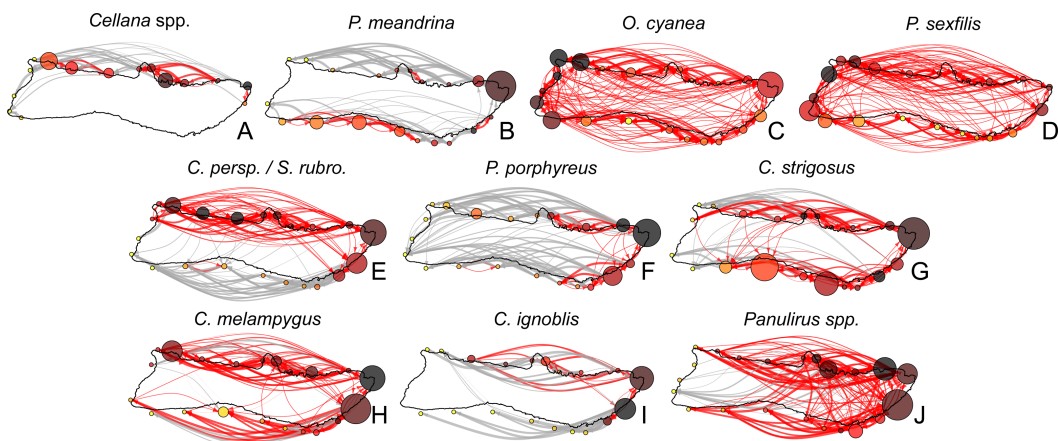

**Figure 7  Moloka'i connectivity networks by species.** Graph-theoretic networks between regions around Moloka'i by species arranged in order of minimum pelagic larval duration. (A–D) Short dispersers (3–25 days), (E–G) medium dispersers (30–50 days), and (H–J) long dispersers (140–270 days). Node size reflects betweenness centrality of each region, scaled per species for visibility. Node color reflects out-degree of each region; yellow nodes have a low out-degree, red nodes have a medium out-degree, and black nodes have a high out-degree. Red edges are connections in a strongly connected component, while gray edges are not part of a strongly connected component (although may still represent substantial connections). Edge thickness represents log-transformed proportion of dispersal along that edge.

showed fewer strongly connected regions on the south shore than the north shore, with the exception of *C. strigosus*. *P. porphyreus* showed more strongly connected regions east of Kalaupapa but lower connectivity on the western half of the island.

Region-level networks showed both species-specific and species-wide patterns of connectivity (Fig. 8). With a few exceptions, sites along the eastern coast—notably, Cape Halawa and Pauwalu Harbor—showed relatively high betweenness centrality, and may therefore act as multigenerational pathways between north-shore and south-shore populations. In *Cellana* spp.*,* Leinapapio Point and Mokio Point had the highest BC, while in high-connectivity *O. cyanea* and *P. sexfilis,* regions on the west coast had high BC scores. *P. meandrina* and *C. strigosus* showed several regions along the south shore with high BC. For *Cellana* spp. and *P. meandrina,* regions in the northeast had the highest out-degree, and therefore seeded the greatest number of other sites with larvae (Fig. 8). Correspondingly, regions in the northwest (and southwest in the case of *P. meandrina*) showed the highest in-degree. For *O. cyanea* and *P. sexfilis*, regions on the western and southern coasts showed the highest out-degree. For most species, both out-degree and in-degree were generally highest on the northern and eastern coasts, suggesting higher connectivity in these areas.

Several species-wide hotspots of local retention emerged, particularly East Kalaupapa Peninsula/Leinaopapio Point, the northeast point of Moloka'i, and the middle of the south shore. Some species also showed some degree of local retention west of Kalaupapa Peninsula. While local retention was observed in the long-dispersing *Caranx* spp. and *Panulirus* spp., this amount was essentially negligible. In terms of source–sink dynamics, Ki'oko'o, Pu'ukaoku Point, and West Kalaupapa Peninsula, all on the north shore, were the only sites that consistently acted as a net source, exporting more larvae than they import

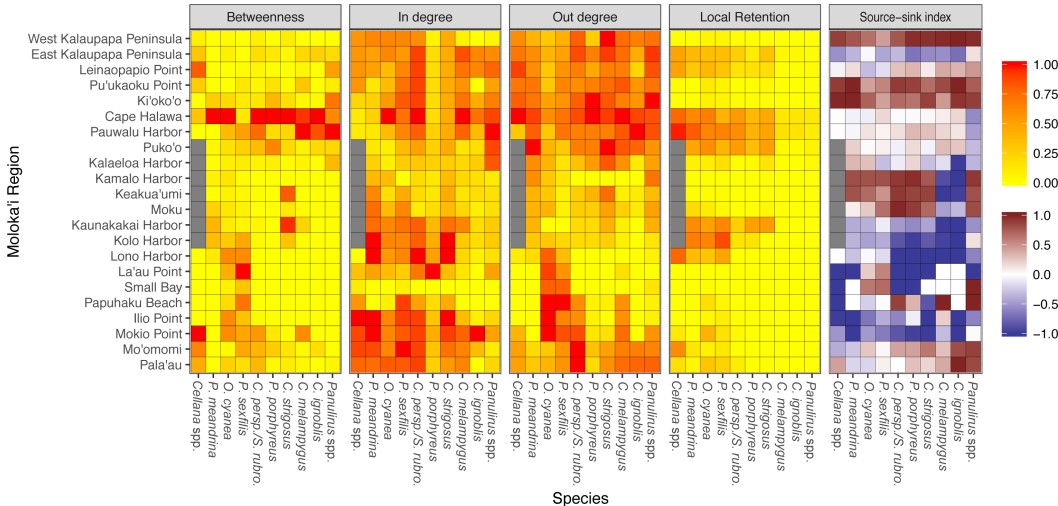

**Figure 8** **Region-level summary statistics across all species.** Betweenness centrality is a measure of the number of paths that pass through a certain region; a high score suggests potentially important multi-generation connectivity pathways. In-degree and out-degree refer to the amount of a node's incoming and outgoing connections. Betweenness centrality, in-degree, and out-degree have all been normalized to values between 0 to 1 per species. Local retention is measured as the proportion of larvae that settled back to their spawn site out of all larvae spawned at that site. Source-sink index is a measure of net export or import; negative values (blue) indicate a net larval sink, while positive values (red) indicate a net larval source. White indicates that a site is neither a strong source nor sink. Gray values for *Cellana* spp. denote a lack of suitable habitat sites in that particular region.

(Fig. 8). Kaunakakai Harbor, Lono Harbor, and Mokio Point acted as net sinks across all species. Puko'o, Pauwalu Harbor, and Cape Halawa were either weak net sources or neither sources nor sinks, which corresponds to the high levels of local retention observed at these sites. Pala'au and Mo'omomi acted as either weak sinks or sources for short dispersers and as sources for long dispersers.

Only four networks showed regional cut-nodes, or nodes that, if removed, break a network into multiple weakly-connected components (Fig. S5). *Cellana* spp. showed two cut-nodes: Mokio Point in northwest Moloka'i and La'au Point in southwest Moloka'i, which if removed isolated Small Bay and Lono Harbor, respectively. *C. perspicillatus,* and *S. rubroviolaceus* showed a similar pattern in regards to Mokio Point; removal of this node isolated Small Bay in this species as well. In *C. ignoblis,* loss of Pauwalu Harbor isolated Lono Harbor, and loss of Pala'au isolated Ilio Point on the northern coast. Finally, in *Panulirus* spp., loss of Leinaopapio Point isolated Papuhaku Beach, since Leinapapio Point was the only larval source from Moloka'i for Papuhaku Beach in this species.

## The role of Kalaupapa Peninsula in inter- and intra-island connectivity

Our model suggests that Kalaupapa National Historical Park may play a role in inter-island connectivity, especially in terms of long-distance dispersal. Out of all regions on Moloka'i, East Kalaupapa Peninsula was the single largest exporter of larvae to Hawai'i Island, accounting for 19% of all larvae transported from Moloka'i to this island; West Kalaupapa

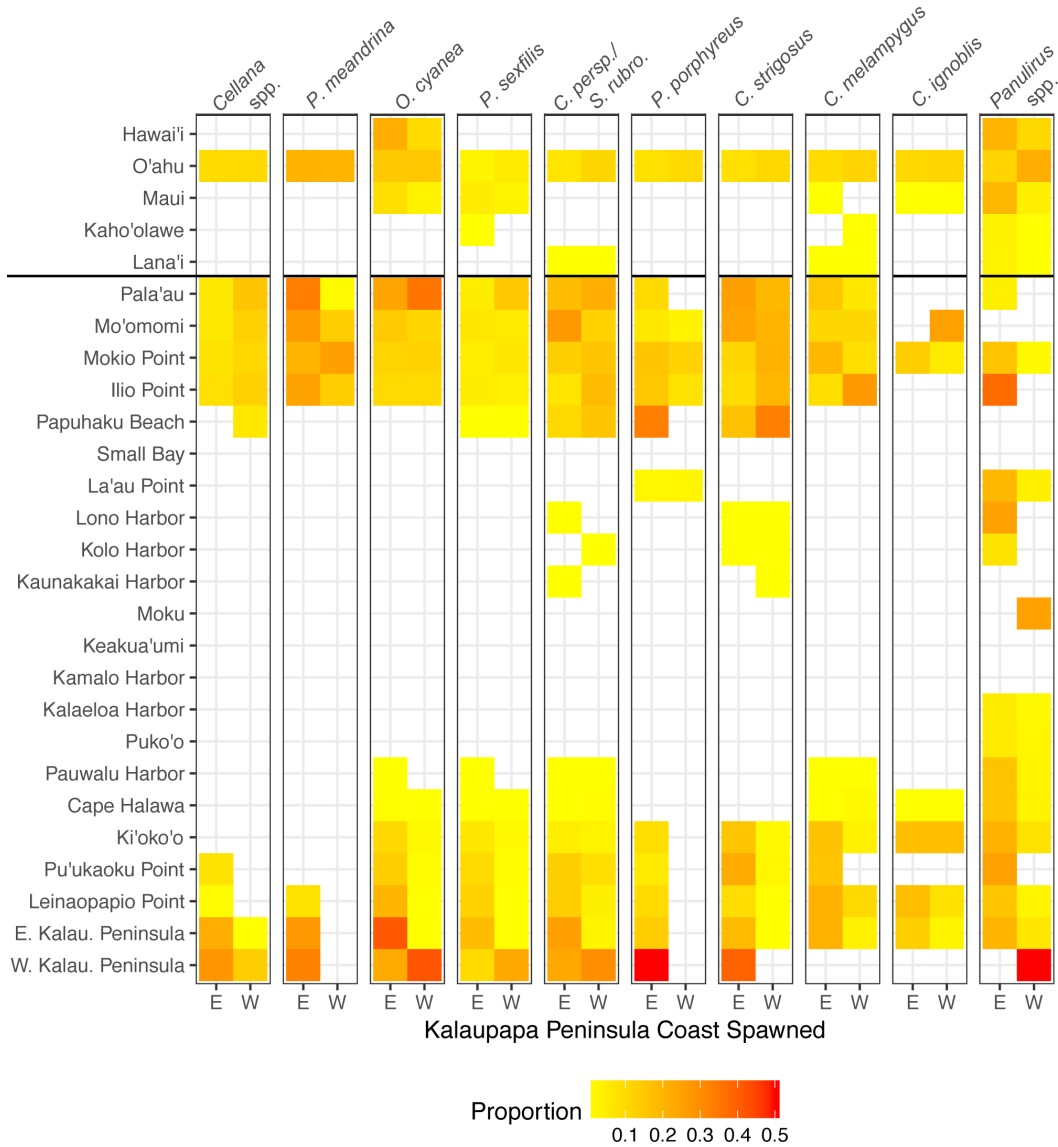

**Figure 9** **Connectivity matrix for larvae spawned on Kalaupapa Peninsula.** Includes larvae settled on Molokaí (regions below horizontal black line) and those settled on other islands (regions above horizontal black line), spawned from either the east (E) or west (W) coast of Kalaupapa. Heatmap colors represent rearward proportion, calculated by the number of larvae settling at site $s$ from site $o$ divided by all settled larvae at site $s$. White squares indicate no dispersal along this path.

Peninsula accounted for another 10%. The park also contributed 22% of all larvae exported from Moloka'i to O'ahu, and successfully exported a smaller percentage of larvae to Maui, Lana'i, and Kaho'olawe (Fig. 9). Kalaupapa was not marked as a cut-node for any species, meaning that full population breaks are not predicted in the case of habitat or population loss in this area. Nevertheless, in our model Kalaupapa exported larvae to multiple regions along the north shore in all species, as well as regions along the east, south, and/or west shores in most species networks (Figs. 9 and 10). The park may play a particularly

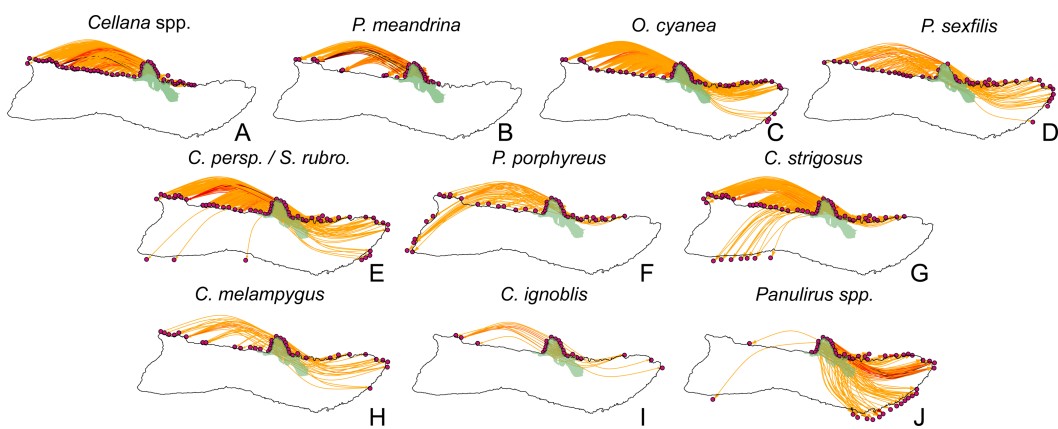

**Figure 10 Larval spillover from Kalaupapa National Historical Park.** Site-level dispersal to sites around Moloka'i from sites in the Kalaupapa National Historical Park protected area, by species. (A–D) Short dispersers (3–25 days), (E–G) medium dispersers (30–50 days), and (H–J) long dispersers (140–270 days). Edge color reflects proportion of dispersal along that edge; red indicates higher proportion while yellow indicates lower proportion. Kalaupapa National Historical Park is highlighted in light green.

important role for long-dispersing species; settlement from Kalaupapa made up 18%–29% of all successful settlement in *Caranx* spp. and *Panulirus* spp., despite making up only 12% of spawning sites included in the model. In *C. strigosus, S. rubroviolaceus,* and *C. strigosus,* Kalaupapa showed a particularly high out-degree, or number of outgoing connections to other regions, and West Kalaupapa was also one of the few regions on Moloka'i that acted as a net larval source across all species (Fig. 8). Our study has also demonstrated that different regions of a marine protected area can potentially perform different roles, even in a small MPA such as Kalaupapa. Across species, the east coast of Kalaupapa showed a significantly higher betweenness centrality than the west ($p = 0.028$), while the west coast of Kalauapapa showed a significantly higher source–sink index than the east ($p = 2.63e{-}9$).

## DISCUSSION

### Effects of biological and physical parameters on connectivity

We incorporated the distribution of suitable habitat, variable reproduction, variable PLD, and ontogenetic changes in swimming ability and empirical vertical distributions of larvae into our model to increase biological realism, and assess how such traits impact predictions of larval dispersal. The *Wong-Ala et al. (2018)* IBM provides a highly flexible model framework that can easily be modified to incorporate either additional species-specific data or entirely new biological traits. In this study, we included specific spawning seasons for all species, as well as spawning by moon phase for *Cellana* spp., *P. meandrina,* and *C. ignoblis* because such data was available for these species. It proved difficult to obtain the necessary biological information to parameterize the model, but as more data about life history and larval behavior become available, such information can be easily added for these species and others. Some potential additions to future iterations of the model might include density of reproductive-age adults within each habitat patch, temperature-dependent

pelagic larval duration (*Houde, 1989*), ontogenetic-dependent behavioral changes such as orientation and diel vertical migration (*Fiksen et al., 2007*; *Paris, Chérubin & Cowen, 2007*), pre-competency period, and larval habitat preferences as such information becomes available.

In this study, we have demonstrated that patterns of fine-scale connectivity around Molokaʻi are largely species-specific and can vary with life history traits, even in species with identical pelagic larval duration. For example, the parrotfish *S. rubroviolaceus* and *C. perspicillatus* show greater connectivity along the northern coast, while the goatfish *P. porphyreus* shows higher connectivity along the eastern half of the island. These species have similar PLD windows, but vary in dispersal depth and spawning season. Spawning season and timing altered patterns of inter-island dispersal (Fig. 5) as well as overall settlement success, which was slightly higher in species that spawned by moon phase (Fig. 2). While maximum PLD did appear play a role in the probability of rare long-distance dispersal, minimum PLD appears to be the main driver of average dispersal distance (Fig. 2). Overall, species with a shorter minimum PLD had higher settlement success, shorter mean dispersal distance, higher local retention, and higher local connectivity as measured by the amount and size of strongly connected components.

The interaction of biological and oceanographic factors also influenced connectivity patterns. Because mesoscale current patterns can vary substantially over the course of the year, the timing of spawning for certain species may be critical for estimating settlement (*Wren et al., 2016*; *Wong-Ala et al., 2018*). Intermittent ocean processes may influence the probability of local retention versus long-distance dispersal; a large proportion of larvae settled to Oʻahu, which is somewhat surprising given that in order to settle from Molokaʻi to Oʻahu, larvae must cross the Kaiwi Channel (approx. 40 km). However, the intermittent presence of mesoscale gyres may act as a stabilizing pathway across the channel, sweeping larvae up either the windward or leeward coast of Oʻahu depending on spawning site. Likewise, in our model long-distance dispersal to Hawaiʻi Island was possible at certain times of the year due to a gyre to the north of Maui; larvae were transported from Kalaupapa to this gyre, where they were carried to the northeast shore of Hawaiʻi (Fig. S6). Preliminary analysis also suggests that distribution of larval depth influenced edge directionality and size of connected components (Fig. 7); surface currents are variable and primarily wind-driven, giving positively-buoyant larvae different patterns of dispersal than species that disperse deeper in the water column (Fig. S7).

## Model limitations and future perspectives

Our findings have several caveats. Because fine-scale density estimates are not available for our species of interest around Molokaʻi, we assumed that fecundity is equivalent at all sites. This simplification may lead us to under- or over-estimate the strength of connections between sites. Lack of adequate data also necessitated estimation or extrapolation from congener information for larval traits such as larval dispersal depth and PLD. Since it is difficult if not impossible to identify larvae to the species level without genetic analysis, we used genus-level larval distribution data (*Boehlert & Mundy, 1996*), or lacking that, an estimate of 50–100 m as a depth layer that is generally more enriched with larvae (*Boehlert,*

*Watson & Sun, 1992*; *Wren & Kobayashi, 2016*). We also estimated PLD in several cases using congener-level data (see Table 1). While specificity is ideal for making informed management decisions about a certain species, past sensitivity analysis has shown that variation in PLD length does not greatly impact patterns of dispersal in species with a PLD of >40 days (*Wren & Kobayashi, 2016*).

Although our MITgcm current model shows annual consistency, it only spans two and a half years chosen as neutral state 'average' ocean conditions. It does not span any El Niño or La Niña (ENSO) events, which cause wide-scale sea-surface temperature anomalies and may therefore affect patterns of connectivity during these years. El Niño can have a particularly strong impact on coral reproduction, since the warm currents associated with these events can lead to severe temperature stress (*Glynn & D'Croz, 1990*; *Wood et al., 2016*). While there has been little study to date on the effects of ENSO on fine-scale connectivity, previous work has demonstrated increased variability during these events. For example, *Wood et al. (2016)* showed a decrease in eastward Pacific dispersal during El Niño years, but an increase in westward dispersal, and *Treml et al. (2008)* showed unique connections in the West Pacific as well as an increase in connectivity during El Niño. While these effects are difficult to predict, especially at such a small scale, additional model years would increase confidence in long-term connectivity estimations. Additionally, with a temporal resolution of 24 h, we could not adequately address the role of tides on dispersal, and therefore did not include them in the MITgcm. *Storlazzi et al. (2017)* showed that tidal forces did affect larval dispersal in Maui Nui, underlining the importance of including both fine-scale, short-duration models and coarser-scale, long-duration models in final management decisions.

We also limit our model's scope geographically. Our goal was to determine whether we could resolve predictive patterns at this scale relevant to management. Interpretation of connectivity output can be biased by spatial resolution of the ocean model, since complex coastal processes can be smoothed and therefore impact larval trajectories. To limit this bias, we focused mainly on coastal and regional connectivity on scales greater than the current resolution. We also used the finest-scale current products available for our study area, and our results show general agreement with similar studies of the region that use a coarser resolution (*Wren & Kobayashi, 2016*) and a finer resolution (*Storlazzi et al., 2017*). Also, while knowledge of island-scale connectivity is important for local management, it does disregard potential connections from other islands. In our calculations of edge density, betweenness centrality and source-sink index, we included only settlement to Moloka'i, discarding exogenous sinks that would bias our analysis. Likewise, we cannot predict the proportion of larvae settling to other islands that originated from Moloka'i, or the proportion of larvae on Moloka'i that originated from other islands.

It is also important to note scale in relation to measures of connectivity; we expect that long-dispersing species such as *Caranx* spp. and *Panulirus* spp. will show much higher measures of connectivity when measured across the whole archipelago as opposed to a single island. The cut-nodes observed in these species may not actually break up populations on a large scale due to this inter-island connectivity. Nevertheless, cut-nodes in species with short- and medium-length PLD may indeed mark important habitat locations, especially

in terms of providing links between two otherwise disconnected coasts. It may be that for certain species or certain regions, stock replenishment relies on larval import from other islands, underscoring the importance of MPA selection for population maintenance in the archipelago as a whole.

## Implications for management

Clearly, there is no single management approach that encompasses the breadth of life history and behavior differences that impact patterns of larval dispersal and connectivity (*Toonen et al., 2011*; *Holstein, Paris & Mumby, 2014*). The spatial, temporal, and species-specific variability suggested by our model stresses the need for multi-scale management, specifically tailored to local and regional connectivity patterns and the suite of target species. Even on such a small scale, different regions around the island of Moloka'i can play very different roles in the greater pattern of connectivity (Fig. 8); sites along the west coast, for example, showed fewer ingoing and outgoing connections than sites on the north coast, and therefore may be more at risk of isolation. Seasonal variation should also be taken into account, as mesoscale current patterns (and resulting connectivity patterns) vary over the course of a year. Our model suggests species-specific temporal patterns of settlement (Fig. 5); even in the year-round spawner *O. cyanea,* local retention to Moloka'i as well as settlement to O'ahu was maximized in spring and early summer, while settlement to other islands mostly occurred in late summer and fall.

Regions that show similar network dynamics may benefit from similar management strategies. Areas that act as larval sources either by proportion of larvae (high source–sink index) or number of sites (high out-degree) should receive management consideration. On Moloka'i, across all species in our study, these sources fell mostly on the northern and eastern coasts. Maintenance of these areas is especially important for downstream areas that depend on upstream populations for a source of larvae, such as those with a low source–sink index, low in-degree, and/or low local retention. Across species, regions with the highest betweenness centrality scores fell mainly in the northeast (Cape Halawa and Pauwalu Harbor). These areas should receive consideration as potentially important intergenerational pathways, particularly as a means of connecting north-coast and south-coast populations, which showed a lack of connectivity both in total number of connections (edge density) and proportion of larvae. Both of these connectivity measures were included because edge density includes all connections, even those with a very small proportion of larvae, and may therefore include rare dispersal events that are of little relevance to managers. Additionally, edge density comparisons between networks should be viewed with the caveat that these networks do not necessarily have the same number of nodes. Nevertheless, both edge density and proportion show very similar patterns, and include both demographically-relevant common connections as well as rare connections that could influence genetic connectivity.

Management that seeks to establish a resilient network of spatially managed areas should also consider the preservation of both weakly-connected and strongly-connected components, as removal of key cut-nodes (Fig. S5) breaks up a network. Sites within a SCC have more direct connections and therefore may be more resilient to local population

loss. Care should be taken to preserve breeding populations at larval sources, connectivity pathways, and cut-nodes within a SCC, since without these key sites the network can fragment into multiple independent SCCs instead of a single stable network. This practice may be especially important for species for which we estimate multiple small SCCs, such as *Cellana* spp. or *P. meandrina*.

Kalaupapa Peninsula emerged as an important site in Moloka'i population connectivity, acting as a larval source for other regions around the island. The Park seeded areas along the north shore in all species, and also exported larvae to sites along the east and west shores in all species except *P. meandrina* and *Cellana* spp. Additionally, it was a larval source for sites along the south shore in the fishes *C. perspicillatus, S. rubroviolaceus,* and *C. strigosus* as well as *Panulirus* spp. Western Kalaupapa Peninsula was one of only three regions included in the analysis (the others being Ki'oko'o and Pu'ukaoku Point, also on the north shore) that acted as a net larval source across all species. Eastern Kalaupapa Peninsula was particularly highly connected, and was part of a strongly connected component in every species. The Park also emerged as a potential point of connection to adjacent islands, particularly to O'ahu and Hawai'i. Expanding the spatial scale of our model will further elucidate Kalaupapa's role in the greater pattern of inter-island connectivity.

In addition to biophysical modeling, genetic analyses can be used to identify persistent population structure of relevance to managers (*Cowen et al., 2000*; *Casey, Jardim & Martinsohn, 2016*). Our finding that exchange among islands is generally low in species with a short- to medium-length PLD agrees with population genetic analyses of marine species in the Hawaiian Islands (*Bird et al., 2007*; *Rivera et al., 2011*; *Toonen et al., 2011*; *Concepcion, Baums & Toonen, 2014*). On a finer scale, we predict some level of shoreline-specific population structure for most species included in the study (Fig. 6). Unfortunately, genetic analyses to date have been performed over too broad a scale to effectively compare to these fine-scale connectivity predictions around Moloka'i or even among locations on adjacent islands. These model results justify such small scale genetic analyses because there are species, such as the coral *P. meandrina*, for which the model predicts clear separation of north-shore and south-shore populations which should be simple to test using genetic data. To validate these model predictions with this technique, more fine-scale population genetic analyses are needed.

## CONCLUSIONS

The maintenance of demographically connected populations is important for conservation. In this study, we contribute to the growing body of work in biophysical connectivity modeling, focusing on a region and suite of species that are of relevance to resource managers. Furthermore, we demonstrate the value of quantifying fine-scale relationships between habitat sites via graph-theoretic methods. Multispecies network analysis revealed persistent patterns that can help define region-wide practices, as well as species-specific connectivity that merits more individual consideration. We demonstrate that connectivity is influenced not only by PLD, but also by other life-history traits such as spawning season, moon-phase spawning, and ontogenetic changes in larval depth. High local retention of

larvae with a short- or medium-length PLD is consistent with population genetic studies of the area. We also identify regions of management importance, including West Kalaupapa Peninsula, which acts as a consistent larval source across species; East Kalaupapa Peninsula, which is a strongly connected region in every species network, and Pauwalu Harbor/Cape Halawa, which may act as important multigenerational pathways. Connectivity is only one piece of the puzzle of MPA effectiveness, which must also account for reproductive population size, long-term persistence, and post-settlement survival (*Burgess et al., 2014*). That being said, our study provides a quantitative roadmap of potential demographic connectivity, and thus presents an effective tool for estimating current and future patterns of dispersal around Kalaupapa Peninsula and around Moloka'i as a whole.

## ACKNOWLEDGEMENTS

The authors would like to thank Y Jia for MITgcm data, JATK Wong-Ala for work in developing the biophysical model, J Wren for advice concerning biophysical modeling, and E Brown for advice and support. Furthermore, we would like to thank members of the Moloka'i kūpuna and fishing community for their expertise and advice, especially to M Poepoe for help in parameterizing the model. Mahalo.

### Funding

This work was supported by the US National Park Service (Task Agreement P16AC01182). This work was completed during an AIAS-COFUND Fellowship to Anna B. Neuheimer at the Aarhus Institute of Advanced Studies, which receives funding from the Aarhus University Research Foundation (Aarhus Universitets Forskningsfond) and the European Union's Seventh Framework Programme, Marie Curie Actions (grant agreement 609033). The funders had no role in study design, data collection and analysis, decision to publish, or preparation of the manuscript.

### Grant Disclosures

The following grant information was disclosed by the authors:
US National Park Service: P16AC01182.
AIAS-COFUND Fellowship.
Aarhus University Research Foundation.
European Union's Seventh Framework Programme: 609033.

### Competing Interests

Robert J. Toonen is an Academic Editor for PeerJ.

### Author Contributions

- Emily E Conklin conceived and designed the experiments, performed the experiments, analyzed the data, prepared figures and/or tables, authored or reviewed drafts of the paper, approved the final draft.

- Anna B Neuheimer conceived and designed the experiments, contributed reagents/materials/analysis tools, authored or reviewed drafts of the paper, approved the final draft, provided the IBM.
- Robert J. Toonen conceived and designed the experiments, contributed reagents/materials/analysis tools, authored or reviewed drafts of the paper, approved the final draft, provided computing resources.

### Data Availability

Conklin, Emily (2018): Moloka'i modeled larval connectivity. figshare. Dataset. https://doi.org/10.6084/m9.figshare.6726413.v1.

### Supplemental Information

Supplemental information for this article can be found online at http://dx.doi.org/10.7717/peerj.5688#supplemental-information.

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
