# Peer review of "Modeled larval connectivity of a multi-species reef fish and invertebrate assemblage off the coast of Moloka‘i, Hawai‘i"

_PeerJ, doi:10.7717/peerj.5688_

## Round 0.1 · original submission · Major Revisions

The reviews in general are very positive and the suggested revisions should tighten up the manuscript to produce an excellent publication. The revisions may require additional analysis but no additional data collection is needed. Overall, tightening of the material means trimming to improve focus.

The past tense is required in Introduction, Methods, and Discussion. Please remove discussion items from the results section and focus the discussion section with no specific call out for conclusions.

The figures are high quality. However, some of them are unnecessary for the comprehension of the study. For example, Figures 3 and 4 can be combined and Figure 12 moved to the Supporting Information. Some of the captions need to be modified and legend is often missing.

The results section should be trimmed to focus on the main questions of the study. Many of the reviewer comments address this.

I look forward to seeing the revised submission.

Reviewer 1 ·

Basic reporting

Overall, the manuscript is clearly written and provides novel and relevant results for the field of marine dispersal and connectivity, as well as marine management. The manuscript covers a wide range of information, dealing with various species, several methods and considering important biological and physical factors.

With a clearly written introduction, the authors provide a background that is supported by relevant literature in the field.

The structure of the manuscript follows the expected style of a scientific manuscript and contains subtitles that speak for themselves.

The results obtained provide a background against which hypotheses are tested and which may be used by marine recourse managers as aimed for by the authors.

Experimental design

The manuscript covers a topic that is highly relevant and uses methods (Lagrangian particle tracking and graph-theory) that are increasingly used to estimate connectivity at temporal and spatial scales that cannot be covered using standard observational techniques.

The aim of the manuscript is clearly defined, being the simulation of larval dispersal around Molokaʻi and quantifying island-scale connectivity. Although model simulation should be interpreted with caution and provide output of which the interpretation can be biased strongly by the resolution of ocean data, studies of this kind are relevant for management practices as they provide insight on the importance of processes and patterns that cannot be studied otherwise. Output from this study should ideally be combined with high-resolution genetic data as underscored by the authors.

The methods used are described in detail. Some minor comments are provided in the "comments to the authors".

Validity of the findings

As to my knowledge, a similar study has no been conducted in the study area and has been undertaken clearly with the aim of filling a knowledge gap as to help advising coastal and marine management. As stated by the authors, more biological data is needed and future studies should be improved further by including aspects such as temperature-dependent pelagic larval duration. Despite some well-known limitations of IBMs arising from limited or scattered species-specific data, the authors provide a interesting study that may stimulate as well the collection of data that is needed to increase the realism of this type of models,

The authors did a great effort on compiling the required biological data and using high-quality ocean data. Since the interpretation of connectivity output can be biased by the spatial resolution of ocean data, it is worth to pay some attention to this in the Discussion section. Drawing conclusions from this type of simulations should be done with the model resolution in mind, given that complex coastal processes may be smoothed and hence affect connectivity estimates and simulated trajectories.

Conclusions drawn by the authors are all based by the analysis performed and speculation is limited.

Additional comments

Dear authors,

the manuscript is well-written and provides insight on aspects of dispersal and connectivity that are highly relevant for conservation and management practices. The results are novel and were obtained using techniques that are increasingly used to study connectivity and dispersal at spatial scales at which standard observational techniques cannot be adequately used. Also, the area studies is of high

I hereby provide some constructive and minor comments:
- Throughout the text: sometimes 'Fig. X', sometimes 'Fig X' (see the dot)
- LINE 112: Which bathymetry dataset has been used to run the MITgcm simulation?
- LINE 115: It may be useful to provide the resolution also in km, between brackets for example
- LINE 152: Although the condition to define settlement is arbitrary and not always straightforward, I wonder about the background against which the 3-km contour has been chosen?
- LINE 178: Caranx should be italicized;
- LINE 194: It would be valuable to know which aspects of the “surrounding geography” were considered in the pooling phase.
- LINE 290 and Fig. 6: It is interesting to note that the “dip” in all three kernels is close to 30-35 km. I wonder how this could be explained, what the underlying process may be. I apologize if I have missed this in the text.
- LINE 323: replace ‘shows’ by ‘showing’
- LINE 525: there is double comma - [...] & Cowen, 2007),,
- Caption Fig. 1: It may be useful to provide the absolute (rather than a relative) resolution ("longitudinal resolution of 1:50 [...]") as to avoid that readers have to go back and forth to keep track of numbers.
- Caption Fig. 3: species names should be italic
- Figure 9 and Figure 11: species names on x-axis should be italicized

Reviewer 2 ·

Basic reporting

The language is clear and professional throughout. The manuscript is well-referenced. For the most part it follows PeerJ standards; However, there’s a fair amount of discussion in the Results section, and the Discussion section is broken further into Discussion and Conclusion sections. The authors may want to tidy this in accordance with PeerJ standards. I’m unsure if PeerJ allows a “Results/Discussion” section.

There are a few instances where the convention used to identify species changes – the authors should review all use of genus and species, to be sure they are correct.

Figures are of general high quality. I have a few suggestions for specific figures, which I will give in the General Comments.

The raw data is supplied. However, it is given in a single large “connectivity” file, without sufficient metadata or explanation of column headings. This file may not be adequately accessible, and the authors might consider breaking it apart by species. Also, larval trajectory data is not provided, which would be required to assess how the physical environment or depth of a virtual larva may affect study results or connectivity.

Experimental design

This study meets the experimental design standards of PeerJ. There are some clarifications that could be made in the methods, and these are outlined in the General Comments.

Validity of the findings

For the most part the findings of this study appear valid and are adequately discussed. There are a few instances where interpretations of a result should be tempered, especially when discussing recruitment or archipelago-scale connectivity. These are discussed in the General Comments.

Additional comments

a. Major concerns:
i. Local retention vs. Self-recruitment. The authors claim to have estimated levels of self-recruitment using their model, but this is not the case so far as I understand. In order to estimate self-recruitment, total recruitment must be known, and without inputs from adjacent islands, it is not clear that total recruitment is known. Thus, the authors should be estimating local retention, which is the proportion of larvae produced locally that recruit locally. This may be a better indicator of local persistence than self-recruitment, anyway.
Importantly, this issue highlights a larger issue in the manuscript, where the authors do not adequately couch their results/discussion with the acknowledgement that their dispersal experiments/results exist in a larger system of larval dispersal and exchange. This issue may come up multiple times in this review. As another example, on line 366 a paragraph describing larval export uses vague language that could easily be misconstrued to exaggerate the “importance” of Moloka’i larval export to other islands in the archipelago. Because archipelago-wide dispersal was not modeled, statements like, “… 83%... of larvae settling to O’ahu were spawned on the north shore of Moloka’i…” are misleading without clarification that this is a percentage of the total larvae migrating from Moloka’i to O’ahu (not of O’ahu recruitment), which is itself some proportion of total larval exchanges modeled. There are other examples of where the language needs to be refined throughout the text.

ii. Scales of dispersal. The authors acknowledge that the diverse species traits modeled result in species-specifically shaped networks, but I’m not sure the choice or handling of metrics is adequate. For example, when comparing networks, edge density likely requires that the number of possible connections to be equal between the networks. In this study this is not the case, so far as I am aware. Thus, if there are fewer possible connections (fewer habitats in the network), edge density is more likely to be high than in a larger network with more habitats (even if the actual number of connections between habitats of interest is the same between both networks). Thus, as it stands, this is not a great metric – even though the conclusions drawn from the analysis may indeed be sound.
Further, the BC analysis needs to be addressed. We know that different species in the model have a different number of connections to other reefs in the archipelago (and even within Moloka’i different species have different habitat nodes). When making comparisons, BC is only useful if the nodes of the networks are the same. Thus, on line 423, where BC comparisons are discussed, the authors need to be sure that the networks are indeed comparable in this way. In addition, BC will change substantially as habitats are added to the network. Are the exogenous sinks included in the BC analysis? Discussion of BC in this case needs to acknowledge that only local connections are being assessed.

iii. These models are probabilistic, but the probabilities associated with connections aren't really addressed.

b. Other considerations – in the order that they appear in the manuscript:

L135-141: Interesting that land and benthic masks were not extracted from the MITgcm. Not a criticism, but I’m curious how these coastal and benthic masks overlap with the hydrodynamic mask.
L145: Were w velocities in the model? If not, did particles move up and down in the water column, or were they statically associated with a specific depth?
L153: This is also very interesting, and different from many other particle tracking models. I’m concerned with how this random distribution of PLDs interacts with mortality. I understand that mortality was modeled using a half-life, but it seems PLD is randomly distributed. If a particle’s individual PLD is reached before it finds habitat, and before its ‘mortality’, does it die? Does this imply that mortality is higher than stated in the methods?
L158: There doesn’t appear to be a table summarizing the releases for each species. This could be incorporated into Table 1.
L171 and throughout: Review to make sure that Latin names are being handled appropriately. Also, and for example, I believe the use of Cellana spp. sometimes drops the “spp.”.
L245: I believe this information needs to come earlier.
L261: Italicize o and s.
L265: Should it be rearward proportion rather than probability?
L273: I think “longer time to competency” is more appropriate here than “higher PLDs”?
L303: Are recruitment sites on other islands included in this edge-density estimation?
L314: A loud and clear distinction needs to be made that “connectivity” is referring to within island connectivity, right?
L338: I agree with the authors here, and think this statement needs to be made even more obvious, and the concept referred to throughout.
L358-363: The authors make this seem like a small percentage (5% or 7%), but doesn’t this seem like a very large percentage? 7% of total recruitment in the model was interisland exchange with Maui… that seems significant to me! Part of the difficulty is that these aren’t translated into probabilities.
L366: Please address language in this paragraph that is a bit misleading (discussed above).
L383: … (Figure 8)…?
L385-396: This whole paragraph needs to be re-worded to address some sort of “regional” retention, in my opinion. It’s also unclear if these “realized” connections have high probability. This larval dispersal model should be probabilistic, but probabilities are not really addressed in this analysis. Is a very low probability connection “realized”?
- General question - : What happens to deep dispersers if they encounter shallower water?
L403: Vague sentence. Between the north and south coasts? Within them?
L412: “Bigger picture…” also vague.
L473: I think it would be great to highlight the Park on the maps, maybe showing the park boundaries. Another figure to consider is showing spillover connections from the Park.
L504: Again, I think it needs to be made clear that connectivity is within a small domain. Longer PLDs would likely be correlated with higher connectivity in a larger domain.
L525: ,,
L556: This section may not be necessary here. It could be moved to Intro/Background.
L620: I don’t think this claim can be made, at least in this fashion. There’s no evidence it’s an “important” source, although you have shown it is a potential source. There’s no indication what proportion of total recruitment the connectivity you’ve demonstrated would represent, nor what the probability of arrival might be.
Table 1: The “Spawn period” column could be made clearer. Maybe: (D-D/HR-HR). The ordering of superscripts is hard to follow. Could add the total number of releases.
Fig. 1: Possible to indicate or color the island of Moloka’i?
Fig. 4: It’s not a HUGE deal, but it’s a little vague how these regions were chosen and groups made.
Fig. 5: (Note my earlier concerns about edge-density, also retention). Might be helpful to include PLD, time to competency, # particles released, etc. in this figure, maybe up above the results.
Fig. 8: Y-axis – proportion of what?
Fig. 10: (Note concerns about calculating BC). Consider making the arc-direction consistent, so that connectivity is, for example, always clockwise along an arc.
Fig. 11: There’s a duplicate panel (two BC).
Fig. 12: These could be indicated in Fig. 10.
Fig. 13: Consider using Latin names here. The inconsistency makes it difficult to interpret. Also, I suggest labeling the settlement locations as “Moloka’i” and “Other island”, or similar.

Reviewer 3 ·

Basic reporting

- The English language is clear and professional throughout the manuscript. The use of some abbreviations is sometimes excessive (particularly in the figures captions). The past tense is required in Introduction, Methods, and Discussion. The beginning of each section should be indented.

- The structure of the manuscript is conformed to Peerj standards. Some sentences in the Results should be moved to the Discussion (see General comments).

- The introduction and background are showing the context and the literature is relevant. The structure of the introduction needs to be clarified and some information added. Change to past tense.

- The figures are high quality. However, the number of figures is high and some of them are not relevant for the comprehension of the study. For example, Figures 3 and 4 should be combined and Figure 12 moved to the Supporting Information. Some of the captions need to be modified and legend is often missing (see General comments).

- The results section is very large and some of the results are not relevant for the main questions of the study (see General comments).

- The raw data are supplied.

Experimental design

- The research is original and within the scope of Peerj.
- The study was performed rigorously.
- The methods are well described and enough details are given to replicate.

Validity of the findings

The results are meaningful and the data is robust. The authors performed a lot of analyses. However, more focus should be done on the relevant results (see General comments).

Additional comments

Title: precise that your manuscript is about larval connectivity.

Introduction
Lines 40-55: This first paragraph contains good background information but should be reorganized. First describe why studying larval dispersal/connectivity is important for management; next describe the larval dispersal phases.
Lines 59-61: Genetic and microchemistry studies applied to larval dispersal should be mentioned here (see Leis et al 2011).
Line 71: what do you mean by “larval physiological characteristics”?
Lines 70-73: You should also discuss about pre-competency period and fecundity. This sentence should be re-written.
Line 74: Remove “in the form of”
Line 75: The name of the hydrodynamic model without describing it in the introduction is not understandable. Just mention “hydrodynamic” or “oceanographic” model.
Line 75: “…fine-scale patterns of larval exchange…”
Line 103: The use of “Here” is not really nice in a publication. Instead, use “In this study…”
Line 104: “… to quantify the larval exchanges…”
Lines 103-110: You need to better describe the main research questions of your study (i.e., effects of biological factors on larval connectivity, larval supply from Moloka’i, regional connectivity patterns)

Methods
Use past tense in Methods.
Line 113: Your paragraph should start with: “The hydrodynamic model used was…”
Lines 127-129: The potential effects of El Niño or La Niña events on larval connectivity should be more described in the discussion.
Lines 129-131: What did the comparison show?
Line 149: What do you mean by development?
Line 153: Explain why you chose to randomly assign a PLD value instead of taking a mean PLD for example. Did you run simulation with fixed PLD? Have the results changed?
Line 158: “… equating from…”
Line 158: Those numbers should be added in Table 1 for each species.
Lines 159-163: What about larva pre-competency period? Mention if no data was available.
Line 179: and instead of with
Line 179: August-December
Lines 193-194: The regional description of the habitat needs to be clarified. What are the groups 2 to 11 mean? (2 to 11 instead of 2-11).
Lines 244-257: not sure why you are talking about self-recruitment to evaluate source-sink dynamics. Another metric that is important to consider, particularly for self-persistence, is local retention. Why studying self-recruitment?

Results
Authors should re-write the Results section. This section is too long and we lose the flow of the important findings. The authors need to keep the more relevant results for their study. Also, some sentences belong to the Discussion.
Lines 267-343: please reference to the names of each metric from Figure 5. Sometimes, we do not know which metric you are talking about.
Line 282: …4% for?
Lines 300-313: I believe you should refer to Figure 5 here again (same in last paragraph)
Line 344: With a larval dispersal model, you do not have results on recruitment. You need to change recruitment for settlement throughout the manuscript.
Line 347: Is figure 7 about self-recruitment? It looks like you only have settlement information on Figure 5. Please precise and modify if necessary.
Lines 350-356: This statement is for the discussion.
Lines 363-365: Results are discussed in the Discussion. Move this to the Discussion.
Lines 366-374: Which figure shows these results?
Lines 378-383: These statements are for the discussion.
Lines 385-396: Figure 9 is showing density values and you are describing percentages.
Lines 393-396: Discussion.
Lines 397-409: Where do these percentages come from?
Line 411: “(as defined in Figure 4)”
Lines 432-472: You give too much detailed information. You need to focus on the relevant results for your study.

Discussion
Use past tense.
Line 494: “In this study…” instead of Here
Line 499: “depth of larval distribution” instead of larval dispersal depth
Lines 518-526: and pre-competency period?
Lines 536-539: The same sentences are in the Methods. Remove them from here.
Lines 543-546: How the connectivity patterns could be affected? Give examples with references.
Line 555: You should also have a paragraph about the effect of the abiotic factors on your results. Some of that is mentioned in the Results section and can be moved here.
Line 556-566: I don’t understand the pertinence of this section. Graph theory has been applied to larval connectivity in several previous studies. Why don’t you write about the relevant results that are obtained with graph theory here?
Lines 572-577: This should be in the paragraph about effects of biological factors.
Line 584: settlement instead of recruitment
Lines 586-598: This is not related to the topic of this section. Should be moved.
Line 643: Why are you talking about local retention here?
Lines 644-647: These are important results and are unfortunately lost in the large flow of all the results you are presenting. You need to focus on these results throughout your manuscript.

Table
Table 1: DoY? Day of the Year
Figure 1: You are starting the caption by “Displaced…”. You need to make a sentence. Add “and” after “December 31st, 2011,…”. A color legend referring to the velocities values is better than describing that in the caption. You do not really describe this figure in your manuscript. You can put it in the appendix.
Figure 3: Spawning sites.
Figures 3 and 4 could easily be combined. Also in Figure 4, does each color represent a specific habitat? Precise.
Figure 5: L-R?
Also write “divided by” instead of putting /.
“Mean and maximum...”
“Proportion” and “maximum” in the Figure.
Figure 6: It is hard to distinguish between the colors. Do not fill the areas under the curve and put a different color or type of line for each curve.
Figure 9: Specify that the results in this figure are for Moloka’i.
Figure 10: Add in the figure which ones are short/medium/long dispersers. Also avoid abbreviations in caption.
Figure 11: Having a color legend with values is better than describing it in the caption.
Figure 12: This figure does not give a lot of information. It should be in the appendix.

---

## Round 0.2 · Minor Revisions

The edits made have really improved the work and both reviewers now only list minor revisions. I look forward to seeing the final version.

Reviewer 1 ·

Basic reporting

The language used throughout the manuscript is clear and professional. Figures are generally of high quality, and well described. The structure in conform PeerJ standards. However, there are still a number of style inconsistencies and typo’s for which the text should be checked and some aspects may need further clarification or could be rephrased for clarity. I have provided my suggestions and comments in more detail in the ‘comments for authors’.

Experimental design

The study consists of original research and, by answering well-defined questions, provides insight that is relevant for spatial management. The methods are described in much detail. There are some aspects that may need further clarification, such as whether or not the model includes tides. I have provided my suggestions and comments in more detail in the ‘comments for authors’.

Validity of the findings

The results presented in the study appear valid and the authors have discussed the results in much detail. The authors have included an additional paragraph to stress the importance of considering the model resolution when interpreting results from model simulation as used in this study.

Additional comments

Dear authors,

kindly find herewith some detailed comments and suggestions to be considered:

Sometimes X km, sometimes Xkm (spacing); Sometimes X-Xkm, sometimes X – Xkm (hyphen type); Sometimes Xd, sometimes X days

L124-125: “(…) which is designed for the study of 
 dynamical processes in the ocean on a horizontal scale.” The way this is presented feels a bit odd. Consider removing this part of the statement and rephrase as follows: “The hydrodynamic model selected was the MITgcm, which is designed for the study of 
dynamical processes in the ocean on a horizontal scale. This model solves incompressible 
Navier-Stokes equations to describe the motion of viscous fluid on a sphere, discretized using a 
finite-volume technique (Marshall et al., 1997).”

L124-125: I am wondering if the MITgcm solution includes tides? If so, it would be valuable for the reader to explicitly state this, as for this type of systems, tides constitute an important process in shaping simulated coastal and near-shore trajectories, and hence, influence dispersal distances and long distance dispersal potential. If tides were not considered, I think it’d be valuable to include one or two sentences that cover the potential implications of not having considered this process in simulating trajectories and connectivity patterns. Given that the authors have a paragraph on limitations already (L517), the authors could consider to group all information on ‘Model limitations and future perpectives’ in one section.

L158-159: Consider rephrasing as follows: “Vertical velocities (w) were not implemented (…)” to enhance readability.

L160-161: “an Eulerian method” instead of “a Eulerian method”

L160: replace ‘time-step’ by ‘timestep’ for consistency

L194-195: “At each time step, a larva’s depth was checked against bathymetry, and was assigned to the deepest available layer if the species-specific depth was not available at these coordinates.” This statement is not fully clear and would benefit from further clarification. I wonder what the justification is for assigning to the deepest available layer. Is this fully arbitrary of based on field-based knowledge/literature?

L198: “occurs May-July” should be “occurs in May-July”

L234-237: “To calculate (…), by scaling (…).” This sentence seems incomplete. Could rephrase as “We calculated the forward settlement proportion, i.e. the proportion (…), by scaling (…).” Note: ‘the’ forward settlement proportion (L234) and ‘a’ specific settlement site (L235).

L267: This statement seems to define a strongly connected component (SCC) rather than a weakly connected component (WCC). A WCC is a subgraph in which all nodes are not reachable from other nodes. Hence “A weakly connected component (WCC) is a subgraph in which all nodes are not connected, either indirectly or directly. 


L299-302: It is interesting to wonder how the timing of spawning events relative to the semi-diurnal tidal range variations would influence dispersal and connectivity patterns in these species.

L305: “showed the highest XXX at 8.1%”. Kindly consider rephrasing as follows: “Local retention was lowest for Caranx spp. (<1%) and highest for O. cyanea and P. sexfilis (8.1% and 10%, resp.).

L493: “(…) along the eastern half of the island, These (…)”; replace comma by point.

Table 1: This table is very informative! For consistency, consider using same hyphen in all cells.

Reviewer 3 ·

Basic reporting

The authors replied to all the comments and edited their manuscript as suggested by all the reviewers. The manuscript is clear and the main results are highlighted. I recommend this manuscript for publication after some minor reviews.

Experimental design

no comment

Validity of the findings

no comment

Additional comments

Line 33: reveal
Line 93: Replace biophysical model by individual-based model. The individual-based model is coupled to the oceanographic model. Both IBM and oceanographic model define a biophysical model.
Lines 152-161: I noticed that you moved this paragraph from the Discussion. I don’t think it should be located at the end of your Intro. Because you do not mention Graph Theory in the Intro, you should move this paragraph in the “Graph-theoretic analysis” paragraph in the Methods. And maybe you can shorten a bit the paragraph.
Line 1077: IBM instead of “individual-based model”

Figures 3 and 5: On both figures, the x-axis is the same color as one of the PLD length for Figure 3 and location for Figure 5 which is confusing. Just change the x-axis color (black for example).

---

## Round 0.3 · accepted · Accept

I really like this work and I think it is a valuable addition to the body of knowledge. Thanks for choosing PeerJ as the publisher.

# Reviewer 1 ·

Basic reporting

Following previous revisions, I have no further comments.

Experimental design

Following previous revisions, I have no further comments.

Validity of the findings

Following previous revisions, I have no further comments.

Reviewer 3 ·

Basic reporting

The authors have answered to all the comments and edits. I recommend this manuscript for publication.

Experimental design

no comment

Validity of the findings

no comment